



# Comparing SOA volatility distributions derived from isothermal SOA particle evaporation data and FIGAERO-CIMS measurements

Olli-Pekka Tikkanen[1], Angela Buchholz[1], Arttu Ylisirniö[1], Siegfried Schobesberger[1], Annele Virtanen[1] and Taina Yli-Juuti[1]

[1]Department of Applied Physics, University of Eastern Finland, Kuopio, 70210, Finland

*Correspondence to*: Olli-Pekka Tikkanen (op.tikkanen@uef.fi)

**Abstract.**

The volatility distribution of the organic compounds present in SOA at different conditions is a key quantity that has to be
captured in order to describe SOA dynamics accurately. The development of the filter inlet for gases and aerosols (FIGAERO) and its coupling to chemical ionization mass spectrometer (CIMS) has enabled near simultaneous sampling of gas and particle phases of secondary organic aerosol (SOA) through thermal desorption of the particles. The thermal desorption data has recently been shown to be interpretable as a volatility distribution with the use of positive matrix factorization (PMF) method. Similarly, volatility distribution can be inferred from isothermal particle evaporation
experiments, when  the particle size change measurements are analyzed with process modelling techniques. In this study we compare the volatility distributions that are retrieved from FIGAERO-CIMS and particle size change measurements during isothermal particle evaporation with process modelling techniques. We compare the volatility distributions at two different relative humidity (RH) and two oxidation condition. At high RH conditions, where particles are in a liquid state, we show that the volatility distributions derived the two ways are comparable within reasonable assumption of uncertainty in the
effective saturation mass concentrations that are derived from FIGAERO-CIMS data. At dry conditions we demonstrate the volatility distributions are comparable in one oxidation condition and in the other oxidation condition the volatility distribution derived from the PMF analysis shows considerably more high volatility matter than the volatility distribution inferred from particle size change measurements. We also show that the Vogel-Tammann-Fulcher equation together with a recent glass transition temperature parametrization for organic compounds and PMF derived volatility distribution estimate
are consistent with the observed isothermal evaporation under dry conditions within the reported uncertainties. We conclude that the FIGAERO-CIMS measurements analyzed with the PMF method are a promising method for inferring organic compounds' volatility distribution, but care has to be taken when the PMF factors are interpreted. Future process modelling studies about SOA dynamics and properties could benefit from simultaneous FIGAERO-CIMS measurements.



## 1 Introduction

Aerosol particles have varying effects on health, visibility and climate (Stocker et al., 2013). Organic compounds comprise a substantial amount of atmospheric particulate matter (Jimenez et al., 2009; Zhang et al., 2007) of which a major fraction is of secondary origin, i.e., low-volatility organic compounds formed from oxidation reactions between volatile organic compounds (VOCs) and ozone, hydroxyl radicals and nitrate radicals (Hallquist et al., 2009). The aerosol particles containing these kind of oxidation products are called secondary organic aerosols (SOA) as opposed to primary organic

aerosols i.e. organic particles emitted directly to the atmosphere.  VOC oxidation reactions result in thousands of different organic compounds (Goldstein and Galbally, 2007). A recent review by Glasius and Goldstein, (2016) pointed out that our understanding of SOA is still lacking especially on formation and deposition, and their response to different physicochemical properties of the organic compounds such as volatility. In addition, also the phase state of the organic compounds has been shown to play a role in the SOA dynamics (Reid et al., 2018; Shiraiwa et al., 2017; Yli-Juuti et al., 2017; Renbaum-Wolff et

al., 2013; Virtanen et al., 2010)

The physicochemical properties of organic aerosols can be studied directly and indirectly. The Aerodyne Aerosol mass spectrometer (AMS, Canagaratna et al., 2007; DeCarlo et al., 2006; Jayne et al., 2000) enabled direct and online composition measurements of atmospheric particles for the first time. Combining AMS data with statistical dimension reduction

techniques such as factor analysis and positive matrix factorization (PMF; Zhang et al., 2011, 2007, 2005; Paatero and Tapper, 1994) allowed researchers to draw conclusion on sources and types of atmospheric organic particulate matter from the relatively complex mass spectra data.

The chemical ionization mass spectrometer (CIMS; Lee et al., 2014) coupled with the Filter Inlet for Gases and AEROsols

(FIGAERO-CIMS, Lopez-Hilfiker et al., 2014) is a prominent online technique to study both the gas and particle phases of SOA. During particle phase measurements, a key advantage over the AMS is the softer chemical ionization that retains much more of the molecular information of the compound than the electron impact ionization used in the AMS. Typically the collection of the particulate mass is conducted at room temperature which minimises the loss of semi-volatile compounds during collection. In addition to the overall chemical composition, the gradual desorption of the particulate mass from the

FIGAERO filter yields the thermal desorption behaviour of each detected ion, i.e., it is a direct measure of each ion's volatility. FIGAERO-CIMS measurements have been carried out in both laboratory and field environments to study SOA composition from different VOC precursors and in both rural and polluted environments (Breton et al., 2018; Huang et al., 2018; Lee et al., 2018; D'Ambro et al., 2017; Lopez-Hilfiker et al., 2015). However, the volatility information in these data sets have barely been used.






Besides direct mass spectrometer measurements, SOA properties have been inferred indirectly from growth (e.g. Pathak et al., 2007 and references therein)  and isothermal evaporation (Buchholz et al., 2019a; D'Ambro et al., 2018; Yli-Juuti et al., 2017; Wilson et al., 2015; Vaden et al., 2011) measurements. The complexity of the organic compounds in these studies can be alleviated with the use of a volatility basis set (Donahue et al., 2006), where organic compounds are grouped based on
their (effective) saturation concentration. However as, for example, Vaden et al., (2011) and Yli-Juuti et al., (2017) have both shown that the volatility basis sets derived from SOA growth experiments results in too fast SOA evaporation compared to measured evaporation rates when used as input in process models. This raises a need for alternative methods to derive organic aerosol volatility against which the volatilities inferred from the direct particle size measurements can be compared to.


Recently, Buchholz et. al., (2019b) demonstrated that the FIGAERO-CIMS measurements during particle evaporation can be mapped to volatility distribution of organic compounds by conduction a PMF analysis. On the other hand,  Tikkanen et al., (2019) showed that the volatility distribution can be inferred from isothermal particle evaporation measurements by optimizing evaporation model input to match the measured evaporation rate at different humidity conditions. In this study,
we compare these two approaches for varying oxidation and particle water content conditions. Our main research questions are 1) Are the volatility distributions derived from particle size change during isothermal evaporation and from the FIGAERO-CIMS measurements comparable? 2) How to interpret the PMF results of FIGAERO-CIMS data in terms of volatility? 3) Can a recent published glass transition temperature parametrization (DeRieux et al., 2018) combined with the PMF analysis be used to model particle phase mass transfer limitation observed in evaporation at dry conditions, i.e., in the
absence of particle phase water?

## 2 Methods

### 2.1 Experimental particle evaporation data

The experimental data we use is the same as reported in Buchholz et al., (2019a,b). We briefly summarize the measurement setup below. We generated the particles with a Potential Aerosol Mass (PAM) reactor (Kang et al., 2007; Lambe et al., 2011)
from the reaction of α-pinene with $O_3$ and OH at three different oxidation levels (average oxygen-to-carbon O:C ratios of 0.53, 0.69, and 0.96). We focus on the lowest O:C (0.53) and medium O:C (0.69) experiments in this work. We chose a monodisperse particle population (mobility diameter $d_p$ = 80 nm) with two nano tandem type differential mobility analyzers (nano-DMA; TSI inc., Model 3085) from the initial polydisperse particle population. The size selection diluted the gas phase initiating particle evaporation. The monodisperse aerosol was left to evaporate in a 100 L stainless steel residence time
chamber (RTC). We measured the particle size distribution during the evaporation with a scanning mobility particle sizer (SMPS; TSI inc., Model 3082+3775). The RTC filling took approximately 20 minutes and we performed the first size



distribution measurement at the middle of the filling interval. To obtain short residence time data (data before 10 minutes of evaporation) we added a bypass to the RTC which led the sample directly to the SMPS. By changing the length of the bypass tubing, we were able to measure the particle size distribution between 2 s and 160 s of evaporation. We measured the
isothermal evaporation up to 4 – 10 hours depending on the measurement. We performed the measurements for each oxidation level both at high relative humidity (RH = 80%) and at dry conditions (RH < 2 %). The change in particle size with respect to time are called evapograms. In an evapogram, the horizontal axis presents evaporation time and vertical axis shows the evaporation factor (EF), i.e., measured particle diameter divided by the initially selected particle diameter.

To classify the oxidation level of the particles, we derived the average O:C ratio from composition measurements with a High-Resolution Time-Of-Flight Aerosol Mass Spectrometer (AMS, Aerodyne Research Inc.). Further, we conducted detailed particle composition measurements with an Aerodyne Research Inc. FIGAERO, Lopez-Hilfiker et al., 2014) coupled with a chemical ionization mass spectrometer (CIMS) with iodide as the reagent ion (Aerodyne Research Inc., Lee et al., 2014).  In the FIGAERO inlet, particles are first collected on a PTFE filter. Then the collected particulate mass desorbs
slowly due to gradually heated nitrogen flow which is then transported into the CIMS for detection. We derive the average chemical composition of the particles by integrating the detected signal of each ion over the whole desorption interval. For each ion, the change of detected signal with desorption temperature is called thermogram and generally, the temperature at the maximum of the thermogram ($T_{max}$) is correlated to the volatility of the detected ion. Similar to Bannan et al., (2019) and Stark et al., (2017), we calibrated the $T_{max}$ - volatility relationship using compounds with known vapour pressure.


We collected  particle for  FIGAERO-CIMS analysis at two different stages of the evaporation. We refer to these stages as either "fresh" or "RTC" samples. The fresh samples were collected for 30 minutes directly after the selection of the monodisperse population. The RTC samples were collected after 3 to 4 hours of evaporation in the RTC for 75 minutes. The collected particulate mass was 140–260 ng and 20–70 ng for fresh and RTC samples, respectively. More details about
sample collection, desorption parameters, and data analysis can be found in Buchholz et al., (2019a).

### 2.2 The volatility distribution

We represent the myriad of organic compound in the SOA particles with a one-dimensional volatility basis set (1D VBS, below only VBS, Donahue et al., 2006). VBS groups the organic compounds into 'bins' based on their effective (mass) saturation concentration $C^*$, defined as the product of the compounds activity coefficient and saturation concentration.
Generally, a bin in the VBS represents the amount of organic material in the particle and gas phases. In our study, the walls of the RTC have been shown to work as an efficient sink for gaseous organic compounds (Yli-Juuti et al., 2017). Thus, we can assume that the gas phase in our experimental setup does not contain organic compounds, i.e., the amount of organic matter in a bin is the amount in the particle phase. To distinguish from a traditional VBS that groups the organic compounds to bins where there is a decadal difference in $C^*$ between two adjacent bins, we call the VBS in our work a volatility





distribution (VD). We present the amount of material in each VD bin as dry mole fractions, i.e., mole fractions of the organics, excluding water. In the analysis presented below, we assign properties to each VD bin (e.g. molar mass) treating each bin as if it consisted of only a single organic compound with a single set of properties. We label these pseudo-compounds as "VD compounds" to distinguish them from real organic compounds. The physicochemical properties of each VD compound are listed in Table 1 as well as the ambient conditions of each evaporation experiment.

### 130    2.3 Deriving volatility distribution from an evapogram

We followed the similar approach as in Yli-Juuti et al., (2017) and Tikkanen et al., (2019) to derive a VD at the start of the evaporation from an evapogram. To model the evaporation at high RH, we used a process model (liquid-like evaporation model, hereafter LLEVAP) that assumes a liquid-like particle, i.e., a particle where there are no mass transfer limitations inside the particle and where the rate of change of the mass of a VD compound in the particle phase can be calculated

directly from the gas phase concentrations of this VD compound near the particle surface and far away from the particle (Vesala et al., 1997; Lehtinen and Kulmala, 2003; Yli-Juuti et al., 2017). In this case, the main properties defining the evaporation rate are the saturation concentrations of each VD compound and their amount in the particle.

We used the LLEVAP model to characterize the volatility ranges interpretable from the evaporation measurements. We

calculated the limits by modelling evaporation of a hypothetical particle that consists of one organic compound at dry conditions iterating the range of $\log_{10}(C^*)$ values from -5 to 5. We determined the minimum $C^*$ value with "detectable evaporation", i.e., at least 1% change in particle diameter during the evaporation time (up to 6 h) and the maximum $C^*$ value before "complete evaporation" occurred, i.e., 99% evaporation within 10 s. The minimum $\log_{10}(C^*)$ calculated with this method was -3 $\mu gm^{-3}$ and the maximum $\log_{10}(C^*)$ was 2. We then modelled the particle composition with six VD

compounds with $C^*$ values between these minimum and maximum values. Each VD compounds has a decadal difference in $C^*$ to adjacent VD compounds (the traditional VBS). We note that based on this analysis all the compounds with $\log_{10}(C^*) <$ -3 will not evaporate during the experimental time scale. This means that any compounds with lower $C^*$ than this threshold will be assigned to the $\log_{10}(C^*) = -3$ VD compound. Similarly, any compound with $\log_{10}(C^*) > 2$ will be classified into the $\log_{10}(C^*) = 2$ VD compound or not be detected at all due to evaporating almost entirely before the first measurement point.


We calculated the dry particle mole fraction of each VD compound at the start of the evaporation by fitting the evaporation predicted with the process model to the measured evapograms. Our goal was to minimize the mean squared error in vertical direction between the experimental data and the LLEVAP output. We used the Monte Carlo Genetic Algorithm (MCGA, Berkemeier et al., 2017; Tikkanen et al., 2019) for the input optimization. In the optimization, we set the population size to

be 400 candidates, number of elite members to 20 (5% of the population), number of generations to 10, and number of candidates drawn in the Monte Carlo (MC) part to 3420 which corresponds to half of the total process model evaluations



done during the optimization. We performed the optimization 50 times for each evapogram and selected the best fit VD estimate for further analysis.

The VD derived from the evapograms are hereafter referred to as the VD$_{evap}$. The initial composition of the SOA particles in the dry and wet experiments were the same and can be described by the same fitted VD$_{evap}$ as the particles were generated at the same conditions in the PAM and only the evaporation conditions changed.

## 2.4 Deriving volatility distribution from FIGAERO-CIMS measurement

As shown by Bannan et al., (2019) and Stark et al., (2017), the peak desorption temperature, T$_{max}$, can be used together with
careful calibration to link desorption temperatures from the FIGAERO filter to C$^*$ values for the detected ions. In principle, this would allow us to assign one C$^*$ value to each ion thermogram. But this assumes that one detected ion characterized by its exact mass is indeed just one compound. In practice, this is not always the case and for some ion thermograms a bimodal structure or distinct shoulders/broadening are visible. This can be caused by isomers of different volatility which cannot be separated even by high resolution mass spectra.


Another complication arises due to the thermal desorption process delivering the collected aerosol mass into the CIMS. Especially multi-functional, and hence low volatility compounds may thermally decompose before they desorb from the filter, and thus be detected as smaller ions. The apparent desorption temperature is then determined by the thermal stability of the compound and not its volatility. Typically, this decomposition processes start at a minimum temperature and will not
create a well-defined peak shape (Buchholz et al., 2019b, Schobesberger et al., 2018) presumably because an observed decomposition product may have multiple sources, especially when including all isomers, and the ion signal for the respective composition may overlap with the signal of isomers derived from true desorption. E.g., a true constituent of the SOA particle may give rise to an observed main thermogram peak, but it may be broadening and/or tailing if a decomposition product has the same composition. By ignoring this and simply using the T$_{max}$ values, the true volatility of the
SOA particle constituents will be overestimated, i.e., the derived VD will be biased towards higher C$^*$ bins.

To separate the multiple sources possibly contributing to each ion thermogram (isomers and thermal decomposition products), we applied Positive Matrix Factorisation (PMF, Paatero and Tapper, 1994) to the FIGAERO-CIMS data set. PMF is a well-established mathematical technique in atmospheric science mostly used to identify the contribution of different
sources of aerosol particles or trace gases in the atmosphere. PMF represents the measured matrix of time-series of mass spectra, $X$, as a linear combination of a (unknown) number of constant source profiles, $F$, with varying contributions over time, $G$:

$$X = G \cdot F + E \tag{1}$$





*E* is a matrix containing the residuals between the measured (*X*) and the fitted data (*G·F*). Values for *G* and *F* are found by
minimising this residual, *E_{ij}*, scaled by the corresponding measurement error, *S_{ij}*, for each ion *i* at each time j

$$Q = \sum_{i=1}^{m} \sum_{j=1}^{n} \left( \frac{E_{i,j}}{S_{i,j}} \right) \qquad (2)$$

Each row in *F* contains a factor mass spectrum and each column in *G* holds the corresponding time series of contribution by
each factor. In the case of FIGAERO-CIMS data, the time series is equivalent to the desorption temperature ramp during the
thermogram, and will be called "mass loading profile" below. The absolute values (temperature or time) are irrelevant for the
performance of PMF as the "x values" are only used to determine the order of the data points but have no influence on the
model output (Paatero and Tapper, 1994). This allowed us to combine multiple separate thermogram measurements into one
data set and conducting a PMF analysis. This simplified the comparison of factors between measurements. More details
about the PMF method in the specific case of FIGAERO-CIMS data can be found in Buchholz et al., (2019b).

Once the PMF algorithm was applied to the FIGAERO-CIMS data we calculated the VD from the mass loading matrix *G*.
We interpolated each factor's mass loading profile with a resolution of 100 sample points between two temperature steps to
gain sufficient statistics for further analysis. $T_{max}$ was determined as the temperature of the maximum of the factor mass
loading series. We integrated the factor mass loading profile and defined the temperatures where the value of the integral
reaches 25% and 75% of its maximum value. This temperature interval formed the factors desorption temperature range. We
converted the $T_{max}$ value into a $C^*$ value and the desorption temperature range into a $C^*$ range with the parametrization based
derived from calibration measurements with organic compounds with known $C^*$ values.

$$C^* = \frac{\exp\left(\alpha + \beta T_{factor}\right) M_{org}}{R T_{ambient}} 10^9 \qquad (3)$$

where $C^*$ is the effective saturation concentration in units $\mu g m^{-3}$, $M_{org}$ is the molar mass of the organic compound assumed to
be $M_{org} = 0.2$ kg mol$^{-1}$, R is the universal gas constant $T_{factor}$ is the temperature in mass loading profile and $T_{ambient}$ is the
ambient temperature where the evaporation happens (see Table 1), α and β are the fitted coefficients from the calibration
data α=(3.739±0.618) and β=(-0.135±0.009) K$^{-1}$. We applied the lower and higher bounds of the fitting coefficients
uncertainty when we calculated the minimum and maximum for the allowed $C^*$ values in Sect 3.3. Finally, the signal fraction
of each factor was calculated by dividing the integral of a factor's signal over the whole temperature range with the sum of
integrals of all factors' signals. We compare this signal fraction to dry mole fraction in the $VD_{evap}$. We refrained from
converting the counts per second signal into moles as no adequate transmission and sensitivity measurements were available
for the used FIGAERO-CIMS setup. We refer the volatility distribution calculated from the PMF data as $VD_{PMF}$ later in this
work.





With the $T_{max}$ calibration, we can calculate the minimum and maximum $C^*$ values that can be resolved from a FIGAERO

thermogram. The desorption temperature was ramped between 27 °C and 200 °C, but defined peaks (and thus $T_{max}$ values

can be detected only between 30 and 180 °C. Thus, the resolvable $\log_{10}(C^*)$ values range from 1.7 to -11.1. It has to be kept

in mind that strictly this calibration only applies to the $T_{max}$ values of a single ion thermogram.

### 2.5 Modelling particle viscosity at dry conditions

To model the mass transfer limitations observed in the evaporation measurements at dry conditions (Buchholz et al., 2019a)

we used the Kinetic multilayer model for gas particle interactions (KM-GAP; Shiraiwa et al., 2012) with modifications

described in Yli-Juuti et al., (2017) and Tikkanen et al., (2019). The main modifications to the original model was that

during evaporation the topmost layer (the quasi static surface layer) merges with the first bulk layer if the thickness of the

layer is smaller than 0.3 nm. We calculated the viscosity at each layer of the particle as

$$\log_{10}(\eta_j)=\sum_{i=1}^{N} X_{mole,i,j} \log_{10}(b_i),\qquad(4)$$

where $X_{mole,i,j}$ is the mole fraction of the VD compound i in layer j and $b_i$ is a coefficient that describes the contribution of

each VD compound to the overall viscosity.

Since we generated the particles in the same environment (PAM chamber) and only evaporated th at different conditions, the

VD at the start of the evaporation derived from high RH data represents also the composition at the start of the evaporation at

dry conditions. Then we can use the best fit $VD_{evap}$ from the high RH data as input for KM-GAP and fit the $b_i$ values in Eq.

(4) to the dry data set. We set the minimum and maximum allowed values for $b_i$ to $10^{-15}$ and $10^{20}$, respectively. To estimate

the $b_i$ values when modelling the evaporation with $VD_{PMF}$ at dry conditions, we calculated these $b_i$ terms using the mass

spectra of each factor ($F$ in Eq. 1) and the Vogel-Tammann-Fulcher (VTF) equation (DeRieux et al., 2018; Angell, 2002,

1995)


$$\eta_i = \eta_\infty \exp\left(\frac{T_{0,i} D}{T - T_{0,i}}\right),\qquad(5)$$

where $\eta_i$ is the viscosity of a VD compound / PMF factor i which can be seen as a proxy for $b_i$ in an ideal solution, $\eta_\infty$ is the

viscosity at infinite temperature, $T_{0,i}$ is the Vogel temperature of i, and D is a fragility parameter. Setting $\eta_\infty = 10^{-5}$ Pa s and

$\eta(T_g) = 10^{12}$ Pa s (e.g. DeRieux et al., 2018; Gedeon, 2018), where $T_g$ is the glass transition temperature of a compound

yields




$$T_{0,i} \approx \frac{39.14\,T_{g,i}}{39.14 + D}.$$

(6)

We calculated $T_g$ for every compound in the PMF mass spectra with a parametrization for SOA matter developed by DeRieux et al., (2018). We then computed $T_g$ for each PMF factor as a mass fraction weighted sum of glass transition

temperatures of individual compounds (DeRieux et al., 2018; Dette et al., 2014). Based on the $T_{g,i}$ for each PMF factor we calculated the viscosity of each PMF factor with Eqs. (5) and (6) and used them as an approximation for $b_i$. We used fragility parameter value D = 10 according to DeRieux et al., (2018).

## 3 Results

In this section we first focus on the high RH experiments where evaporation is modelled with the LLEVAP model. We will

first compare $VD_{evap}$ and $VD_{PMF}$ when the $C^*$ of a PMF factor is determined from the factor's $T_{max}$. Then, we compare the volatility distributions where the $C^*$ of a PMF factor is determined as the range from the 25[th] and 75[th] percentile desorption temperatures. Lastly, we compare the volatility distributions at dry conditions.

### 3.1 PMF solution interpretation

Figure S1 and Fig S2 show all mass loading profiles derived from FIGAERO-CIMS measurements of evaporation of

medium and low O:C particles at high RH. The corresponding factor mass spectra can be found in Fig. S3 and Fig. S4. A key step in any PMF analysis is determining the "right" number of factors as this can affect the interpretation of the results. A 7-factor solution was chosen for the medium O:C cases and a 9-factor solution for the low O:C ones (see Buchholz 2019b for details). Two additional factors in the low O:C case were needed to capture a contamination on the FIGAERO filter during the dry, fresh sample (factors LC1 and LC2 in Fig. S1 and Fig. S2). As these two factors were clearly an artefact

introduced by the FIGAERO filter sampling, we omitted their contribution from the data set for the following analysis. From careful comparison of the factor profiles and mass spectra with filter blank measurements, we determined that factor MB1 in medium O:C case and factor LB1 in low O:C case describe the filter/instrument background and are thus also excluded from the VD comparison presented below.

Factors 1-5 in both O:C cases exhibit a monomodal peak shape and can thus be characterised by their $T_{max}$ values, factor

MD1 in medium O:C case and factor LD1 in low O:C case needs to be investigated more closely, as its factor mass spectrum and the sometimes bimodal mass loading profile suggest that this factor contains compounds stemming from both direct desorption (desorption T<100 °C) and thermal decomposition (desorption T >100 °C, see Buchholz et al., 2019b for details). To account for this, the factor is split into two with the first half containing the signal from desorption temperature below

100 °C (factor M/LD1a) and the second half containing that above 100 °C (factor M/LD1b). We treat these factors separately. We note that now the latter half of the split factor is dominated by thermal decomposition products so that the





apparent desorption temperature is actually the temperature at which thermal decomposition leads to products which desorb at this temperature. This apparent desorption temperature is thus a lower limit for the decomposing parent compound, i.e., the true volatility of these parent compounds is even lower. However, the desorption temperatures are so high that they lead

to $\log_{10}(C^*) < -3$ and are thus below the comparable range for $VD_{evap}$. Figure 1 (high RH data) and Fig. S5 (dry condition data) show the mass loading profiles derived from FIGAERO-CIMS measurements of medium and low O:C particles' evaporation after we excluded the contamination and background factors and split the decomposition factors.

### 3.2 Volatility distribution comparison at high RH based on factor $T_{max}$

To compare $VD_{evap}$ and $VD_{PMF}$, we need to determine the time interval in the evapogram that the $VD_{PMF}$ represents. We collected the fresh samples directly after the size selection. As the particles were collected on a filter for 30 minutes, the collected sample represents particles that have evaporated from 0 up to 30 minutes in the organic vapour free air. We note that this is different from the standard FIGAERO-CIMS sample collection where particles are collected in a quasi-equilibrium with the surrounding gas phase and no significant evaporation occurs (Lopez-Hilfiker et al., 2014). For RTC

samples, we need to consider also that not all particles have evaporated for the same time due to the filling of the RTC for ca. 20 minutes. We determined the minimum time the particles have evaporated in the RTC as the time when we started the sample collection minus the RTC filling time. We determined the maximum evaporation time in the RTC to be the time when we stopped the sample collection plus the filling time. These minimum and maximum comparison times are shown in Table 2 and they are referred to as minimum and maximum (sample) evaporation time. In addition, we also compare the

volatility distributions at the middle of the sample collection interval, i.e., the mean (sample) evaporation time.

Figure 2 shows $VD_{evap}$ and $VD_{PMF}$ for medium (Fig. 2a-b) and low O:C (Fig. 2c-d) particles at high RH. In the $VD_{PMF}$ calculated from $T_{max}$, values of each factor (black crosses), the factors fall into three different volatility classes within our chosen particle size and experimental time scale: practically non-volatile ($\log_{10}(C^*) \leq -2$, slightly volatile ( $-2 < \log_{10}(C^*) \leq 0,$)

and volatile ($\log_{10}(C^*) > 0$). We use these three volatility classes to compare the volatility distributions in Fig. 3 where each VD compound are grouped to these three volatility classes. Figure 3 presents the $VD_{PMF}$ where $C^*$ of each factor is calculated from the $T_{max}$ value and compares this $VD_{PMF}$ to what $VD_{evap}$ is at the minimum, mean and maximum time FIGAERO samples had evaporated.

After the volatility class grouping is applied, we see that there is an excess amount of matter in the highest volatility class (volatility class 3) of the $VD_{PMF}$ compared to the $VD_{evap}$ in all the cases. With the fresh samples (Fig. 3a and Fig. 3c), the $VD_{PMF}$ seems to be the closest to the $VD_{evap}$ at the very start of the evaporation. With the RTC samples (Fig. 3b and Fig. 3d) the $VD_{PMF}$ does not directly match any of the $VD_{evap}$. For particles with medium O:C, $VD_{PMF}$ shows more contribution of volatility class 2 and less of in volatility class 1 compared to $VD_{evap}$.






To investigate the observed discrepancy more detailed, we used the $VD_{PMF}$ as an input for the LLEVAP model and calculated the corresponding isothermal evaporation behavior (i.e. the evapogram). We show these simulated evapograms in Fig. 4a for the medium O:C case and in Fig 4b for low O:C condition together with the simulated evapogram calculated using $VD_{evap}$ as an input for the LLEVAP model. The derived $VD_{PMF}$ represents the particle composition averaged over the sample collection

interval. To account for this, we run the model by starting the evapogram simulations calculated with $VD_{PMF}$ either at the start of this interval (minimum isothermal evaporation before sample collection), at the mean (mean isothermal evaporation before sample collection), or at the end (maximum isothermal evaporation before sample collection). The simulated evapograms calculated with $VD_{PMF}$ of the fresh samples do not match the measured evapograms, while the evapogram calculated with $VD_{evap}$ agrees well with the experimental evapogram (black lines in Fig. 4), as we expected since this was the

goal of the $VD_{evap}$ determination. If we take the $VD_{PMF}$ of the fresh samples' as the particle composition at minimum evaporation time, the simulated evaporation is slower than the measured evaporation (light blue lines in Fig. 4). If $VD_{PMF}$ is set to be the particle composition at mean or maximum evaporation time the simulated evaporation is faster than the measured one.

Figure 4 shows also the simulated evapograms calculated with $VD_{PMF}$ of the RTC samples (light brown lines in Fig. 4). in these cases, the particles size decreases little within the simulation time scale. With medium O:C particles, the simulated evaporation matches better to the measured evaporation than the simulations calculated with the $VD_{PMF}$ of the fresh sample although the simulated evapograms shows a slightly higher rate of evaporation than what is measured. With low O:C particles, the evaporation calculated with $VD_{PMF}$ is too fast. The shape of the evapogram does not match the measured one.

**3.3 Applying desorption range to characterize the volatility of PMF factors**

The $T_{max}$ value is a practical choice for the characteristic temperature of the desorption process. However, as we saw on Sect. 3.1 the $VD_{PMF}$ calculated from the peak desorption temperatures did not produce the measured evapogram when used as an input to the LLEVAP model. Working under the assumption that all material collected on the FIGAERO filter, including the higher volatility material, is detected in the CIMS and then captured in the PMF analysis we will relax the assumption that

the volatility of the factor is characterized strictly by the $T_{max}$ value of the factor and investigate the $VD_{PMF}$ further. We will explore how the $VD_{PMF}$ changes when the desorption temperature and the resulting $C^*$ are interpreted to contain uncertainty and if the $VD_{PMF}$ considering these uncertainty ranges is consistent with the observed isothermal evaporation. The uncertainty in the desorption temperature raises from the facts that compounds volatilise from the FIGAERO filter throughout the heating and, therefore, one value might not be adequate to characterize the $C^*$ of a factor and that each PMF

factor contains multiple compounds with distinct $C^*$.



We calculated the 25th and 75th percentiles of the desorption temperatures of each factor and converted them to effective saturation concentrations as described in section 2.4 (see diamond markers in Fig. 1). We show the resulting $C^*$ ranges in Fig. 2 as horizontal solid lines where the line colour matches the factor's colour in Fig. 1. We then ran MCGA optimization by
setting a number of compounds equal to the number of PMF factors, molar fraction for each compound at the FIGAERO-CIMS sampling time fixed to the molar fraction of corresponding factor and set the $C^*$ as the optimized variables restricted to the range corresponding to the 25th and 75th percentile desorption temperature. In the optimization the goodness-of-fit statistics was calculated as a mean squared error similar to the determination of $VD_{evap}$.

As the fresh samples were collected between 0 and 30 minutes from the start of the evaporation, we sought for a fitting set of $C^*$ values for for evaporation starting at 0, 15, 30 minutes. Due to scarcity of particle size measurements at collection time of the RTC sample, we will apply this analysis only to the $VD_{PMF}$ of the RTC sample at its minimum evaporation time. In each optimization we set the initial particle diameter to be the same as what is simulated with $VD_{evap}$. We derived 50 $C^*$ estimates for both samples and each evaporation time. From these 50 estimates we chose the best fit evapogram. We refer to these
optimized volatility distributions as $VD_{PMF,opt}$ to separate them from the $VD_{PMF}$ where we used $T_{max}$ to characterize $C^*$ of a PMF factor.

We show the optimized $C^*$ values forming $VD_{PMF,opt}$ in Table 3 for all the studied cases. Figure 5a shows the best fit evaporation simulations calculated with $VD_{PMF,opt}$ of the medium O:C fresh sample. All the simulated evapograms resemble
the experimental evapogram and evapogram calculated with $VD_{evap}$. Figure 5b shows the simulated evapograms calculated with $VD_{PMF,opt}$ for low O:C samples. The evapograms initialized at 0 min and 15 min match the experimental evapogram and the simulated evapogram using $VD_{evap}$ as input. The evapogram using $VD_{PMF,opt}$ starting from the point of 30 minutes of isothermal evaporation does not match the measured evapogram but shows faster evaporation than the measurements.

For finding the $VD_{PMF,opt}$ for the low O:C RTC sample starting at minimum sample evaporation time (168 minutes) we needed to exclude factor LD1a from the calculations to be able to derive the $VD_{PMF,opt}$. As Buchholz et al., (2019b) reported, the mass spectrum of factor LD1a is dominated by compounds that come from the FIGAERO filter / instrument background. In low O:C RTC sample factor L1a is present at such high relative signal strength that its mole fraction is significant to other factors even though the absolute signal strength does not change drastically between the fresh and the RTC sample. The high
relative contribution of factor LD1a is most probably due to the low amount of organic matter available for sample collection.

Overall, these results demonstrate that the fresh and RTC samples can describe the composition of the evaporating particles, when uncertainty in the desorption temperature are considered.





### 3.4 Comparison of the volatility distribution of the fresh and RTC sample at high RH


In this section we compare $VD_{PMF,opt}$ of the fresh samples to $VD_{PMF}$ of the RTC sample to study are the two VD comparable. We compare the two VD at the mean evaporation time of the RTC sample. We calculated the evapograms with $VD_{PMF,opt}$ of the fresh sample starting from different sample evaporation times and recorded the mole fraction of each factor at mean evaporation time of the RTC sample (216 minutes for medium O:C particles and 211 minutes for low O:C particles). Figure

6a and Fig. 6c show this comparison for both medium O:C and low O:C particles where the factors are grouped to the three volatility classes described in Sect. 3.2. To ensure that the factors are grouped to the same volatility classes for each studied VD, we used the $C^*$ values of the $VD_{PMF,opt}$ at mean sample evaporation time as basis according to which the grouping is done.

Assuming that the fresh sample represents particles from the middle of the sampling interval (mean evaporation time), the compositions simulated based on the $VD_{PMF,opt}$ of the fresh samples are comparable to the corresponding $VD_{PMF}$ of the RTC sample in both oxidation conditions. In both O:C levels, the $VD_{PMF,opt}$ of minimum fresh sample evaporation time show higher contribution of volatility class 2 and lower contribution of volatility class 1 than the $VD_{PMF}$ of the RTC sample. Contrary, the $VD_{PMF,opt}$ of maximum fresh sample evaporation time in the medium O:C case (Fig. 6a) shows higher

contribution of volatility class 1 and lower contribution of volatility class 2 than the $VD_{PMF}$ of the RTC sample. These results show that even though we calculated the $VD_{PMF,opt}$ starting from the minimum and maximum possible sample evaporation time, the $VD_{PMF,opt}$ of the fresh samples are consistent with the RTC samples only if the $VD_{PMF,opt}$ represents particle composition around the middle of the sample collection interval.

### 3.5 Volatility distribution comparison at dry condition

Next, we analysed the evaporation experiments under dry conditions where the evaporation rate was reduced compared to the high RH conditions. We interpreted this difference as an indication of particle phase diffusion limitations at dry conditions (Yli-Juuti et al., 2017). Using the initial particle composition information obtained from the high RH experiments and the FIGAERO-CIMS data, we explored the effect of particle viscosity on the evaporation process.

First, we investigated the range of particle viscosities that are required to explain the observed slower evaporation at dry conditions. For this, we simulated the particle evaporation in dry conditions based only on the evapogram data. We used the $VD_{evap}$ (i.e., the initial particle composition obtained by optimizing mole fractions of VD compounds with respect to the observed evapogram at high RH) as the initial condition for the simulations and optimized the $b_i$ values (Eq. 3) for each VD compound. The best fit simulation from this optimization agrees well with the observed size decrease in the dry experiments

for both low and medium O:C particles (Fig. 8, black line). Based on these simulations the viscosity of the particles need to





increase from below $10^5$ Pa s to approximately $10^8$ Pa s during the evaporation in order to explain the evaporation rate observed for the dry particles.

Second, we tested the performance of the composition dependent viscosity parameterization by DeRiuex et al. (2018) used

together with the PMF results. For this, we calculated the volatility distribution, $VD_{PMF,dry}$, based on the $T_{max}$ values of the factors from the fresh sample of the evaporation experiment at dry conditions (in the same way as $VD_{PMF}$ for the high RH case). The mole fraction of each factor was calculated from the mass loading profile giving the initial mole fraction of each VD compound for the simulations. We assigned this $VD_{PMF,dry}$ as the particle composition at the mean evaporation time of the fresh sample, i.e. 15 minutes, and simulated the particle evaporation from there onwards. The particle size at the beginning

of the simulation (i.e. at 15 minutes of evaporation) was taken from the above simulations optimized based only on the evapogram data, which fitted well with the measurements. We calculated the viscosity parameter $b_i$ value for each VD compound as described in Section 2.5 based on the mass spectra of the factor and the parameterization by DeRieux et al. (2018). This resulted in too high viscosity for particles to evaporate in practise at all during the length of the experiment for both low and medium O:C particles (black dashed line in Fig. 8). Therefore, we also made a simulation where the viscosity

parameter $b_i$ value for each factor was calculated based on the viscosity parameterization by setting the $T_g$ values of all compounds 30 K lower than the parametrization predicted, which is in line with the uncertainties reported by DeRiuex et al. (2018). In this case the simulated evaporation was faster than observed (grey dashed line in Fig. 8). This suggest that the observed evaporation rate at dry conditions and the viscosity parametrization by DeRieux et al. (2018) may be consistent with each other within the uncertainty range of the viscosity parametrization and the uncertainty range of the $C^*$ of PMF

factors.

Similar to Fig. 3, we show in Fig. 7 the comparison of $VD_{PMF,dry}$ ($C^*$ from $T_{max}$) to the $VD_{evap}$ at dry conditions with the VD compounds grouped into the three volatility classes. We show the mass loading profiles and the volatility distributions at dry conditions in Fig. S5 and Fig. S6. For medium O:C particles, $VD_{PMF,dry}$ calculated both from fresh and RTC sample have

slightly more contribution of volatility classes 2 and 3 and less of volatility class 1 compared to the corresponding $VD_{evap}$. For low O:C particles, the $VD_{PMF,dry}$ differs substantially from the $VD_{evap}$: considerably more matter is in the highest volatility class (class 3) than in the lowest volatility class (class 1) especially in the case of the fresh sample. Overall, the $VD_{PMF,dry}$ suggests higher volatility compared to the $VD_{evap}$. Therefore, the underestimation of the evaporation rate when using the $VD_{PMF,dry}$ together with the viscosity parameterization (black dashed line in Fig. 8) originates from the high estimated

viscosity.

As a third investigation on the viscosity, we used again the PMF results of the fresh sample at dry conditions to initialize the particle composition in the model at the mean fresh sample evaporation time, i.e., at 15 minutes. Also at this time, the mole fraction of each factor were calculated from the mass loading profile giving the initial mole fraction of each VD compound



for the simulations. Then, using the MCGA algorithm together with the KM-GAP model, we estimated the $b_i$ coefficient and $C^*$ of each VD compound by optimizing the KM-GAP simulated evapogram to the measured evapogram at dry condition. This way we obtained both the initial volatility distribution ($VD_{PMF,dry,opt}$) and viscosity parameters $b_i$ simultaneously. For this optimization, we restricted the $C^*$ values of the factors based on the 25th and 75th percentile of the desorption temperature of the factors, similarly as done above for $VD_{PMF,opt}$, and the viscosity parameter $b_i$ values based on the DeRieux et al. (2018)

parameterization. The $b_i$ values calculated with the original parametrization by DeRieux et al., (2018) were set as the upper limit for $b_i$ values. The lower limit for $b_i$ values were calculated by setting the glass transition temperature of each compound 30 K lower than the parametrization predicted. As above, also in these simulations the initial particle size was taken from the simulations where optimization was based on only the evapogram data. For medium O:C particles it was possible to find a set of $C^*$ and $b_i$ values that produced an equally good match to experimental data as the $VD_{evap}$ produced (purple line in Fig.

8a). For low O:C particles, the match to experimental data was slightly weaker than with the $VD_{evap}$ (yellow line in Fig. 8b).

Figure 6b shows the comparison of the measured and simulated particle composition, on the basis of the three volatility classes, at RTC sample collection time for the dry experiments for low and medium O:C particles. The measured composition is the VD calculated from PMF results of RTC sample at dry conditions and the optimized $C^*$ values of the

factors from the corresponding dry experiment were used for these VD. The simulated particle composition is taken from the optimized model run (optimized $VD_{PMF,opt,dry}$ and $b_i$) at the mean RTC sample collection time similar to the high RH cases presented in Fig. 6a and Fig. 6c. For medium O:C particles the measured and simulated composition at mean of the RTC collection time are in agreement. For low O:C particles there is a clear discrepancy: the measurements imply a much larger relative contribution from the volatility class 1 and a smaller contribution from the volatility class 2 compared to the

simulations. This inconsistency may be related to the rather high viscosities in the simulations. The viscosity of the low O:C particles in this optimized simulation was rather high, $\eta > 10^8$ Pa s, throughout the evaporation, slowing the evaporation of the higher volatility compounds. Similar evaporation curve could be obtained with lower viscosity and lower volatilities of the compounds.

## 4 Discussion

$VD_{PMF}$ and $VD_{PMF,dry}$ capture qualitatively the evaporation dynamics well in all studied cases. For the $VD_{PMF}$ of the fresh samples, the first and second factor desorb at low heating temperatures (below 100 °C) indicating that these factors represent organic compounds that evaporate almost completely from the particles in the experimental time scale of our isothermal evaporation experiments. In the RTC samples, these factors show significantly lower or non-existing signal strength relative to the other factors. The factors that desorb at high temperatures show increase in the relative signal strength in the RTC

samples compared to the fresh samples which is consistent with the expected increase in relative contribution of lower volatility compounds along evaporation.

At high RH, the $VD_{PMF}$ that was derived from $T_{max}$ of each factors mass loading profile did not produce evapogram similar to the measured ones, when the $VD_{PMF}$ was used as an input to the LLEVAP model. This reflects the sensitivity of particle evaporation to the $C^*$ values and suggest that the $VD_{PMF}$ is not directly applicable as a particle composition estimate for detailed particle dynamics study. When we allowed uncertainty in the $C^*$ values of each factor we were able to explain most of the discrepancy between the simulated and measured evapograms. The simulated evapograms, after optimizing the $C^*$ of each factor from their appropriate ranges, are close to the experimental values at all other cases except in the low O:C case when the $VD_{PMF}$ is interpreted to represent particles at the end of fresh sample's collection interval (maximum evaporation time).

Even though we assumed a quite large uncertainty range for the desorption temperature of each factor, the resulting $C^*$ estimates range in most cases around one order of magnitude. In the cases where the $C^*$ range is higher and a factor has high enough signal, the estimated $C^*$ values in $VD_{PMF,opt}$ are closer to the $C^*$ calculated from $T_{max}$ of the factor than the extremes of the range (e.g. factor 2 and factor 4 in medium O:C high RH experiments). This highlights the fact that even though the $C^*$ estimated from $T_{max}$ did not produce exactly comparable evapograms, the $C^*$ values that produce correct evaporation dynamics are not far away from those derived from the $T_{max}$ values.

We note that care has to be taken when PMF results are transferred to volatility distributions, especially with regard to separating the contribution of instrument background and contamination from the true sample. When the sample mass was low (in the low O:C RTC sample) we noticed that the first half of the bimodal (factor LD1a) resulted in a high mole fraction even though the absolute signal strength of the factor did not change between the fresh and the RTC sample, which is usually an indication that this signal is caused by instrument background. Factor LD1a affects the VD calculations only when the collected mass is low. Removing this factor from the low O:C RTC sample allowed us to derive $VD_{PMF,opt}$ that produces an evapogram similar to the experiment. The signal strength of this factor was low enough in all other cases to not affect the overall VD estimation.

$VD_{PMF,dry}$ of the fresh sample in low O:C case showed noticeably higher amount of high volatility matter than $VD_{evap}$. We cannot explain these differences with a single factor like in the low O:C high RH RTC sample case since in dry conditions multiple high volatility factors show up in the PMF solution. This discrepancy between the volatility distributions is not expected and raises a need for further studies on the role of viscosity and possible particle phase chemistry to SOA particle dynamics. Future studies should investigate the possibility of chemical reactions that modify the volatility of organic compounds and how viscosity is described in process models.

## 5 Conclusions

We compared volatility distributions derived from FIGAERO-CIMS measurements with PMF analysis to volatility distributions derived from fitting a process model to match measured size change of particles during isothermal evaporation. In all studied experimental data sets we were able to capture the measured evaporation with the fitting method. With high RH experiments, $VD_{PMF}$ deviated from $VD_{evap}$ especially when the FIGAERO samples were collected at the early stages of the evaporation. However, qualitatively, both types of VD evolved similarly, i.e., the fraction of lower volatility compounds

increased, and the fraction of higher volatility compounds decreased during the particles' evaporation.

The volatility distribution from PMF at high RH matched the experimental values better when we interpreted the volatility of each factor as a range of possible $C^*$ values and optimized the $C^*$ values from these ranges with respect to the measurements.

At dry conditions, we were able to simulate the evapograms based on the PMF results using the VTF equation and glass transition temperature parametrization of DeRieux et al., (2018) when both $C^*$ and viscosity parameters where optimized and allowed to contain reasonable uncertainties. For medium O:C particles also the simulated composition evolution was consistent with the measurements. However, for low O:C particles the measured composition at the later stages of evaporation suggested considerably higher volatility than the simulations.


Based on our analysis we conclude that using the PMF method with FIGAERO-CIMS thermogram data is good estimating the volatility distribution of organic aerosols when the organic compounds present in the particle phase have low volatilities with respect to the sample collection and analysis time scale. Specifically, $VD_{PMF}$ is useful for extracting information about organic compounds that do not evaporate during the evaporation measurements at room temperature. $VD_{PMF}$ is applicable to

detailed particle dynamics studies when desorption temperature of the factor is characterized with a range around the $T_{max}$ value. Furthermore, combining $VD_{PMF,opt}$ with detailed process modelling and input optimization could allow quantification of other physical or chemical properties of organic aerosols since the FIGAERO-CIMS data constrains the particle composition and effectively decreases the search space that needs to be explored with global optimization methods.

*Code availability:* The process models and the version of the MCGA used in this study can be acquired upon request from the corresponding author.

*Author contributions:* OPT, AB, SS, AV and TYJ designed the study. OPT did the calculations with support from AB and TYJ, except for the PMF calculations which were done by AB. AY developed the method to calculate $C^*$ from desorption

temperature with support from SS. All authors participated in the interpretation of the data. OPT wrote the paper with contributions from all co-authors.





*Acknowledgments:* The authors would like to thank Claudia Mohr and Wei Huang for the use of the FIGAERO instrument from the Karlsruhe Institute of Technology and their support during the FIGAERO-CIMS data analysis. Furthermore, we

want to acknowledge Andrew Lambe and Aerodyne Research Inc. for lending us a Potential Aerosol Mass reactor.

*Financial support:* This work was supported by the Academy of Finland Center of Excellence programme (grant no. 307331), the Academy of Finland (project nos. 299544 and 310682), European Research Council (ERC StG QAPPA 335478) and the University of Eastern Finland Doctoral Program in Environmental Physics, Health and Biology.


*Competing interests:* The authors declare that they have no conflict of interest.



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



**Table 1:** The ambient conditions and properties of the organic compounds used in estimating the VD$_{evap}$. The variables are, from top to bottom, temperature (T) during the evaporation, relative humidity (RH), gas phase diffusion coefficient (D$_{g,org}$), molar mass (M), particle phase density (ρ), particle surface tension (σ) and mass accommodation coefficient (α).

| Variable | Medium O:C High RH | Low O:C High RH | Medium O:C dry | Low O:C dry |
|---|---|---|---|---|
| T (K) | 293.85 | 293.75 | 293.75 | 293.35 |
| RH (%) | 82.4 | 83.5 | 0 | 0 |
| D$^a_{gas}$ (cm$^2$ s$^{-1}$) | 0.05 | 0.05 | 0.05 | 0.05 |
| M (g mol$^{-1}$) | 200 | 200 | 200 | 200 |
| ρ (kg m$^{-3}$) | 1200 | 1200 | 1200 | 1200 |
| σ (mN m$^{-1}$) | 40 | 40 | 40 | 40 |
| α | 1 | 1 | 1 | 1 |

[a] The gas phase diffusion coefficients are scaled to correct temperatures by multiplying with a factor of $(T/273.15)^{1.75}$ (Reid et al., 1987)





**Table 2:** Minimum, mean and maximum time that the particles have evaporated during the FIGAERO sample collection. All times are relative to the start of RTC filling.

| Sample | Minimum evaporation time (min) | Mean evaporation time (min) | Maximum evaporation time (min) |
|---|---|---|---|
| Medium O:C high RH fresh | 0 | 15 | 30 |
| Medium O:C high RH RTC | 173 | 216 | 259 |
| Medium O:C dry fresh | 0 | 15 | 30 |
| Medium O:C dry RTC | 170 | 213 | 256 |
| Low O:C high RH fresh | 0 | 15 | 30 |
| Low O:C high RH RTC | 168 | 211 | 254 |
| Low O:C dry fresh | 0 | 15 | 30 |
| Low O:C dry RTC | 152 | 195 | 238 |



**Table 3:** The best fit $C^*$ values for medium O:C and low O:C high RH experiments when $C^*$ values of PMF factors were optimized with respect to the measured isothermal evaporation. For each experiment three different results are given which correspond to simulations initialized with the PMF mole fraction at the minimum, mean and maximum time that the particles have evaporated during the sample collection (See Table 2). The $C^*$ values are rounded to two significant digits and are in
$\mu gm^{-3}$. $C^*$ values below $10^{-3}$ $\mu gm^{-3}$ are not reported explicitly since the evapogram fitting method is not sensitive to these values.

| | Medium O:C fresh sample min evap. time | Medium O:C fresh sample mean evap. time | Medium O:C fresh sample max evap. time | Medium O:C RTC sample min evap time | Low O:C fresh sample min evap. time | Low O:C fresh sample mean evap. time | Low O:C fresh sample max evap. time | Low O:C RTC sample min evap time |
|---|---|---|---|---|---|---|---|---|
| Factor M1/L1 | $9.79 \cdot 10^{-1}$ | $5.36 \cdot 10^{-1}$ | $3.07 \cdot 10^{-1}$ | $3.06 \cdot 10^{-1}$ | $9.76 \cdot 10^{-1}$ | $1.92 \cdot 10^{-1}$ | $1.92 \cdot 10^{-1}$ | $< 10^{-3}$ |
| Factor M2/L2 | 6.10 | $2.32 \cdot 10^{-1}$ | $7.01 \cdot 10^{-2}$ | $9.41 \cdot 10^{-2}$ | 12.61 | $7.98 \cdot 10^{-1}$ | $8.15 \cdot 10^{-1}$ | $1.65 \cdot 10^{-1}$ |
| Factor M3/L3 | $1.68 \cdot 10^{-1}$ | $2.37 \cdot 10^{-2}$ | $9.49 \cdot 10^{-3}$ | $9.50 \cdot 10^{-2}$ | $2.90 \cdot 10^{-1}$ | $2.57 \cdot 10^{-2}$ | $2.48 \cdot 10^{-2}$ | $7.32 \cdot 10^{-2}$ |
| Factor M4/L4 | $1.39 \cdot 10^{-2}$ | $< 10^{-3}$ | $< 10^{-3}$ | $< 10^{-3}$ | $5.52 \cdot 10^{-2}$ | $2.65 \cdot 10^{-3}$ | $2.65 \cdot 10^{-3}$ | $2.14 \cdot 10^{-2}$ |
| Factor M5/L5 | $< 10^{-3}$ | $< 10^{-3}$ | $< 10^{-3}$ | $< 10^{-3}$ | $1.14 \cdot 10^{-2}$ | $1.55 \cdot 10^{-3}$ | $< 10^{-3}$ | $7.33 \cdot 10^{-3}$ |
| Factor D1a | 70.14 | 10.86 | $1.67 \cdot 10^{-2}$ | 11.82 | 59.73 | $7.48 \cdot 10^{-1}$ | $7.34 \cdot 10^{-1}$ | / |
| Factor D1b | $< 10^{-3}$ | $< 10^{-3}$ | $< 10^{-3}$ | $< 10^{-3}$ | $< 10^{-3}$ | $< 10^{-3}$ | $< 10^{-3}$ | $< 10^{-3}$ |





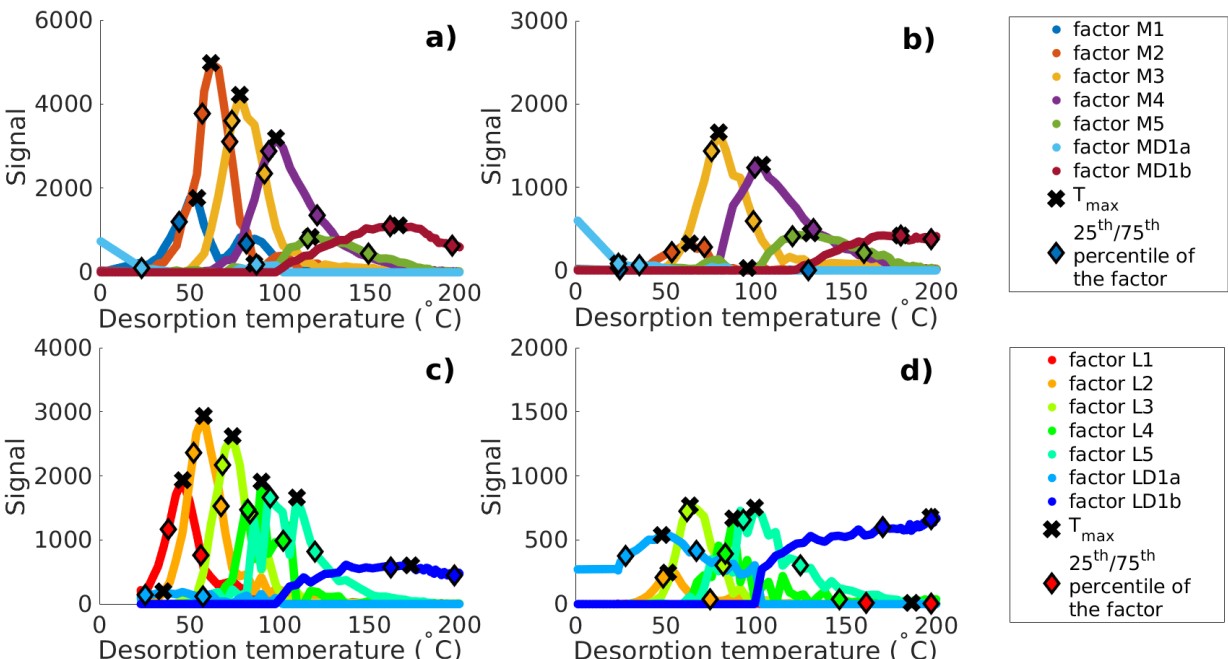

**Figure 1**: Main PMF mass loading profiles for high RH conditions a) medium O:C fresh sample, b) medium O:C RTC sample, c) low O:C fresh sample, d) low O:C RTC sample. Black crosses indicate the peak desorption temperature $T_{max}$ and diamonds mark the 25th and 75th percentiles of the factors area.





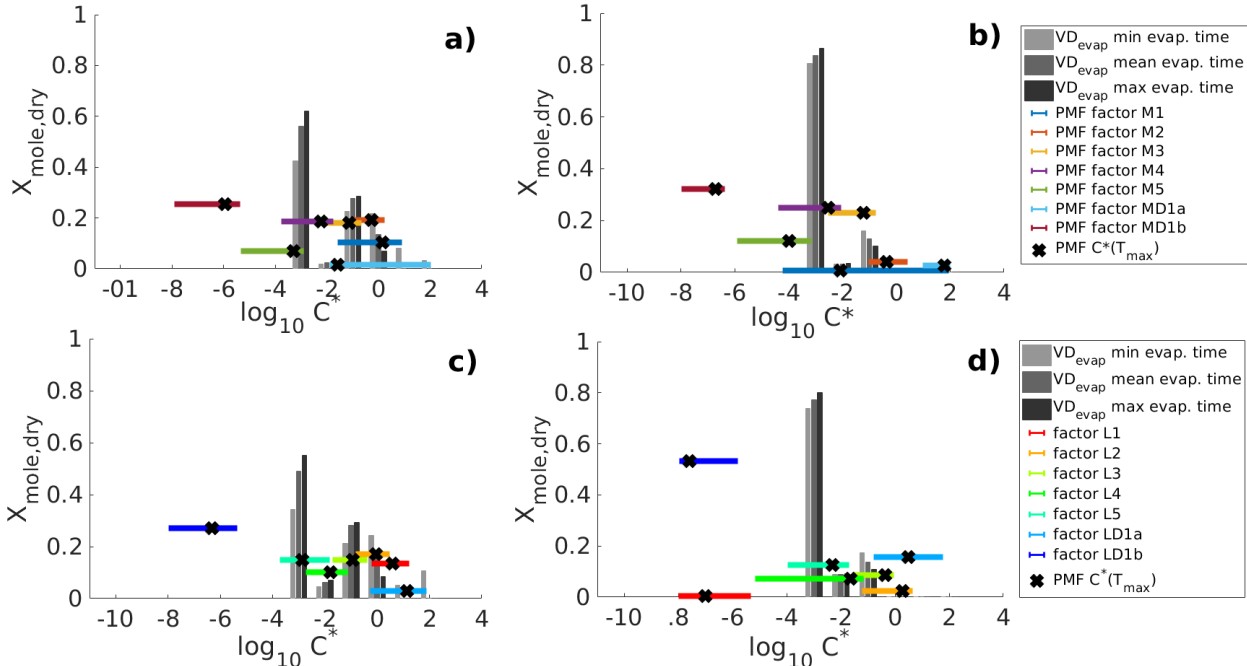

**Figure 2:** Volatility distributions in high RH experiments determined from model fitting (VD$_{evap}$) and PMF analysis (VD$_{PMF}$) on FIGAERO-CIMS data for a) medium O:C fresh sample, b) medium O:C RTC sample, c) low O:C fresh sample, d) low O:C RTC sample. VD$_{evap}$ is shown for the best fit simulation (grey bars). The different grey shades show the VD$_{evap}$ in the simulation at minimum, mean and maximum time that the particles have evaporated when the FIGAERO sample was collected (see Table 2). Black crosses show the $\log_{10}(C^*)$ calculated for each PMF factor from the peak desorption temperature T$_{max}$. The horizontal colored lines show the range of $\log_{10}(C^*)$ calculated from the 25$^{th}$ and 75$^{th}$ percentiles of each PMF factors mass loading profile.





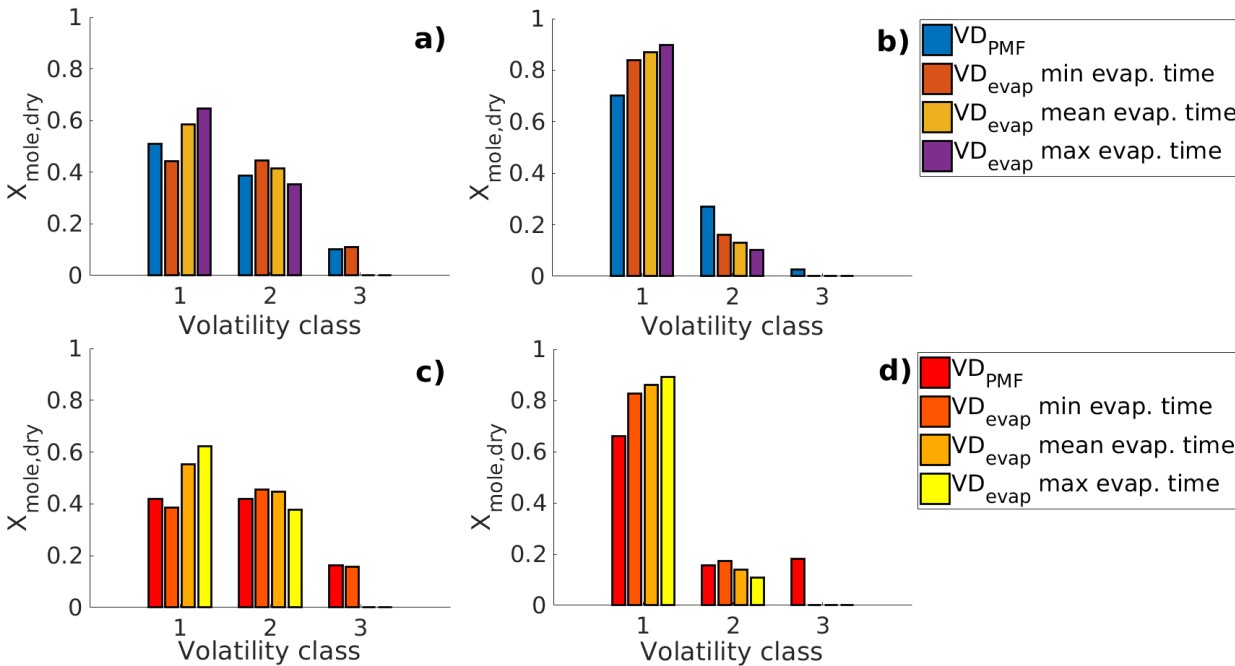

**Figure 3:** Comparison of $VD_{PMF}$ and $VD_{evap}$ in high RH experiments. The VD compounds are grouped into three different volatility classes. Min, mean and max evaporation time refer to the FIGAERO sample collection times presented in Table 2. The volatility classes are 1: $\log(C^*) \leq -2$, 2: $-2 < \log(C^*) < 0$, 3: $\log(C^*) > 0$. a) medium O:C fresh sample, b) medium O:C RTC sample, c) low O:C fresh sample, d) low O:C RTC sample.





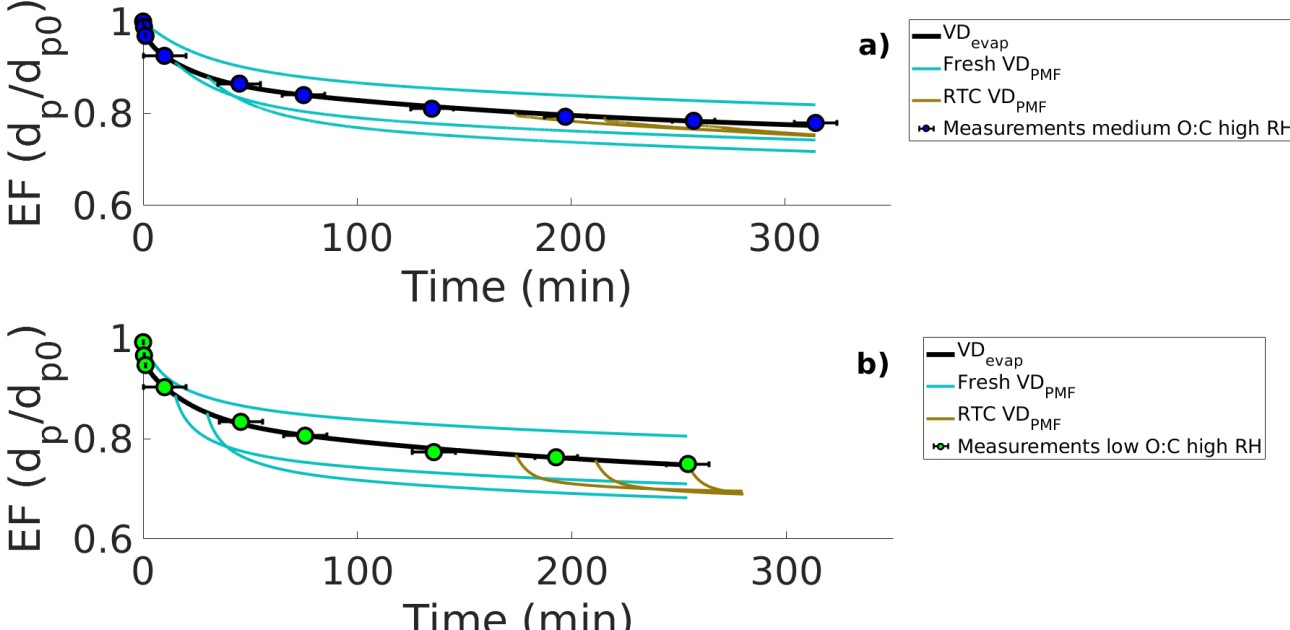

**Figure 4**: Evapograms showing the high RH measured evaporation factors (circles) and their uncertainty in time, LLEVAP simulated evapograms calculated using the best fit $VD_{evap}$ (black solid lines) and LLEVAP simulated evapograms calculated with $VD_{PMF}$ (turquoise solid lines for $VD_{PMF}$ of fresh sample and light brown solid lines for $VD_{PMF}$ RTC sample. a) medium O:C b) low O:C.





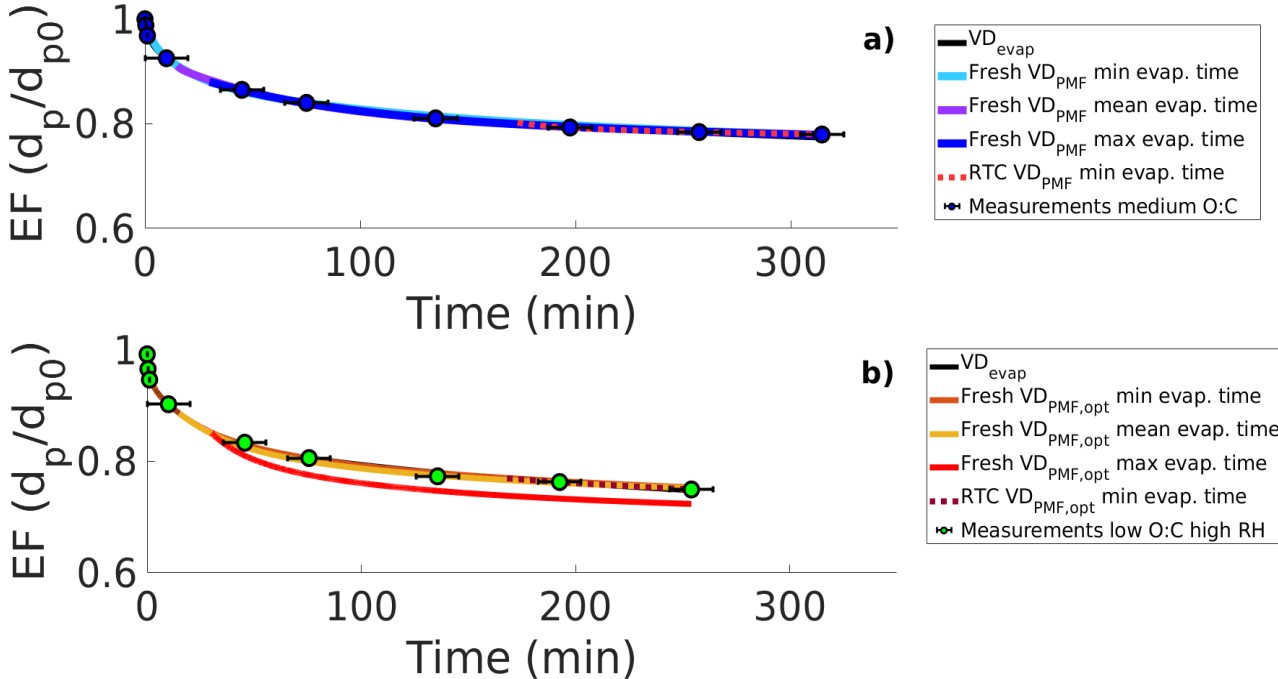

**Figure 5**: Evapograms showing the high RH measured evaporation factors (circles) and their uncertainty in time (black whiskers) and the best fit simulated evapogram calculated with $VD_{evap}$ (black solid line). Other lines show the best fit simulated evapogram calculated with $VD_{PMF,opt}$. a) medium O:C, b) low O:C.



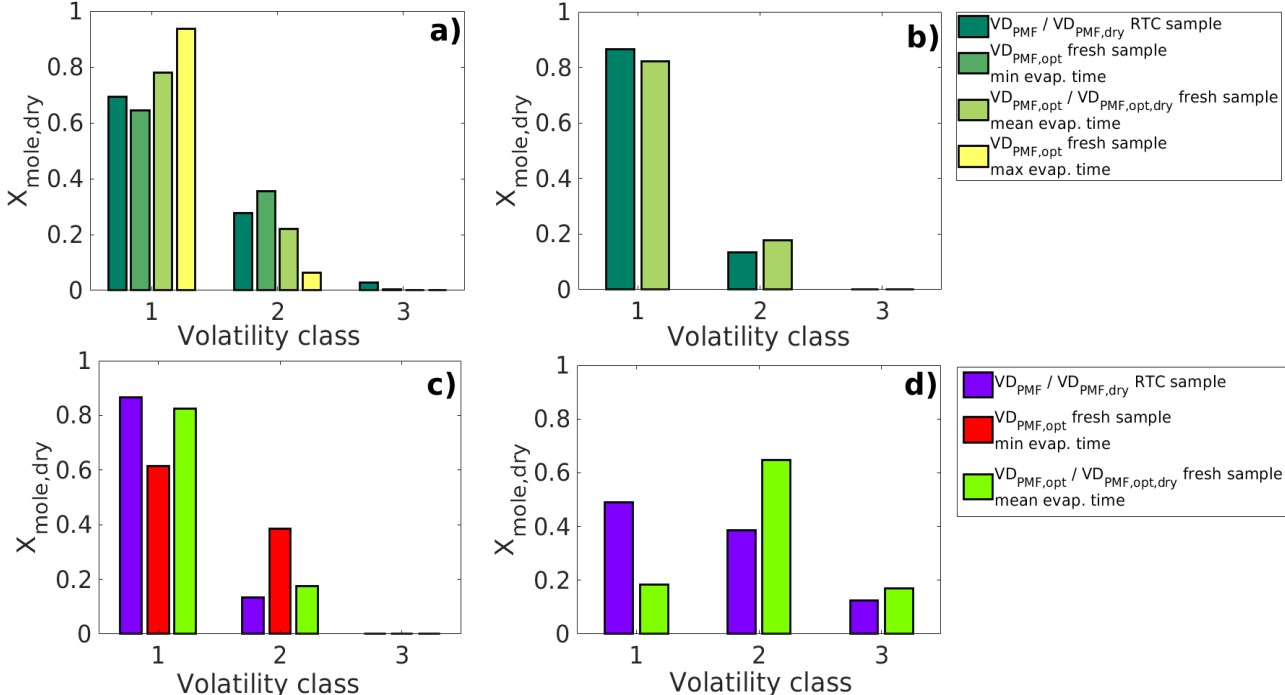

**Figure 6**: Comparison of $VD_{PMF,opt}$ ($VD_{PMF,opt,dry}$ for subfigures b and c) of the fresh samples to $VD_{PMF}$ ($VD_{PMF,dry}$ for subfigures b and c) of RTC samples at the mean time of the collection interval of the RTC sample. The subscripts min, mean and max refer to the points from the fresh sample collection interval from where the $VD_{PMF,opt}$ simulations were initialised (see Table 2). The volatility classes are 1: $\log(C^*) \leq -2$, 2: $-2 < \log(C^*) < 0$, 3: $\log(C^*) > 0$. a) medium OC high RH samples, b) medium O:C low RH samples, c) low O:C high RH samples d) low O:C low RH samples.





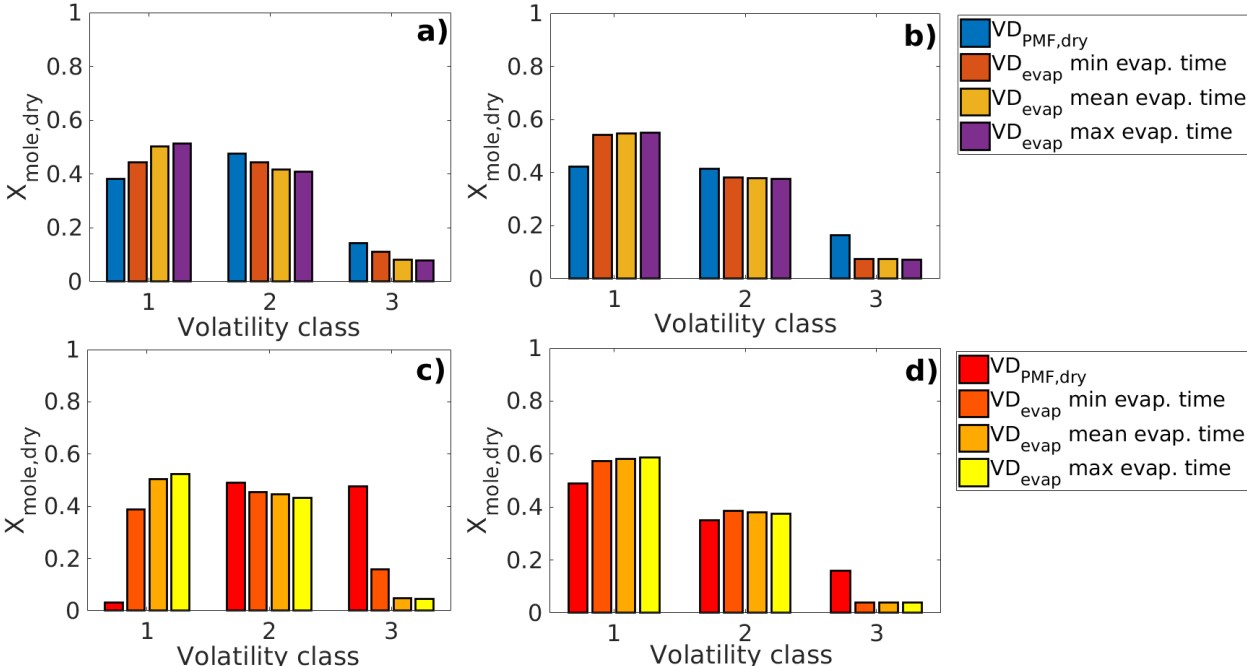

**Figure 7**: Comparison of $VD_{PMF,dry}$ and $VD_{evap}$ in dry condition experiments where the VD compounds are grouped into three volatility classes. Min, mean and max evaporation time refer to the FIGAERO sample collection times presented in Table 2. The volatility classes are 1: $log(C^*) \leq -2$, 2: $-2 < log(C^*) < 0$, 3: $log(C^*) > 0$. a) medium O:C fresh sample, b) medium O:C RTC sample, c) low O:C fresh sample, d) low O:C RTC sample.



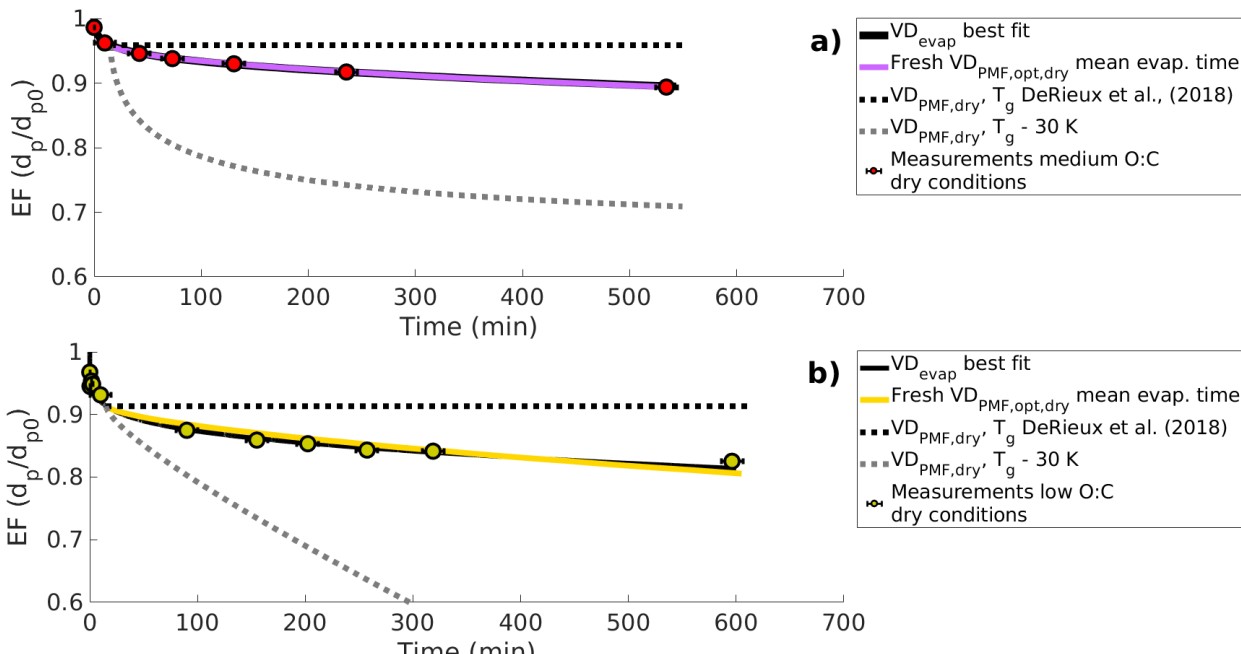

**Figure 8:** Evapogram showing the measured isothermal evaporation of a) medium O:C particles b) low O:C particles at dry conditions and their uncertainty in time (red and yellow markers and black whiskers) and the best fit simulated evapogram calculated with $VD_{evap}$ (black solid line). Purple and yellow solid lines show the best fit simulated evapograms calculated with $VD_{PMF,opt,dry}$ assuming that the FIGAERO sample represents particles at mean of the sample collection interval (see Table 2). Black and grey dashed lines show the $VD_{PMF,dry}$ simulated evapograms where particle viscosity is calculated using the VTF equation and glass transition temperature $T_g$ according to DeRieux et al., (2018) (black dashed line) or $T_g$ is calculated according DeRieux et al., (2018) and 30 K is subtracted from the $T_g$ value (grey dashed line).