# Peer review of "Comparing SOA volatility distributions derived from isothermal SOA particle evaporation data and FIGAERO-CIMS measurements"

_Atmospheric Chemistry and Physics, 2019_

## Referee Comment (RC1) · Anonymous Referee #1 · 19 Dec 2019

In this manuscript, Tikkanen et al. compare SOA volatility derived from two different analyses: isothermal evaporation data and PMF applied to FIGAERO-CIMS data. Volatility is a critical property of organic aerosol, and I agree with the authors that the volatility data from FIGAERO-CIMS measurements have been under-utilized. This manuscript focuses on the comparison of the FIGAERO-CIMS PMF volatility data to isothermal evaporation data; the details of the FIGAERO-CIMS PMF volatility analysis are described in another manuscript currently under review (Buchholz et al., 2019 https://www.atmos-chem-phys-discuss.net/acp-2019-926/). It seems to me that this manuscript can only be accepted if Buchholz et al., 2019 is also accepted. I also request that my specific comments below be addressed before publication of this

[Figure]

manuscript.

1. The authors use data from previously conducted experiments in which SOA was formed from alpha-pinene and ozone/OH. By varying experimental conditions, the SOA produced had low, medium or high O:C. Here, the authors only use experiments when the SOA formed had low or medium O:C. This choice is currently not explained or justified in the manuscript. In my opinion, the dataset utilized here is fairly limited, and the analysis would benefit from inclusion of these additional high O:C data. For example, overall the authors find that the agreement between the two volatility analyses is better for intermediate O:C than for low O:C, and I am curious about the agreement at high O:C.

2. The main takeaways from the manuscript should be clarified. The authors state in the abstract that "FIGAERO-CIMS measurements analyzed with the PMF method are a promising method for inferring organic compounds' volatility distribution". The more detailed results point to the method working better under some conditions than others. It would be useful if the authors could make more concrete recommendations for future use of this method (PMF applied to FIGAERO-CIMS data) to obtain information on organic aerosol volatility.

Editorial comments: There are several typographical and grammatical errors in the manuscript. I include a list of examples below:

Line 19: "volatility distributions derived the two ways are comparable within reasonable assumption"

Line 233: "and only evaporated th at different conditions"

Line 311: "To investigate the observed discrepancy more detailed"

Line 376: "In this section we compare VDPMF,opt of the fresh samples to VDPMF of the RTC sample to study are the two VD comparable."

Line 521-522: "thermogram data is good estimating the volatility distribution of organic

**[ACPD](ACPD)**

Interactive
comment

aerosols"

---

## Referee Comment (RC2) · Anonymous Referee #2 · 28 Jan 2020

Review of acp-2019-927" Comparing SOA volatility distributions derived from isothermal SOA particle evaporation data and FIGAERO-CIMS measurements" by Olli-Pekka Tikkanen, Angela Buchholz, Arttu Ylisirniö, Siegfried Schobesberger, Annele Virtanen, and Taina Yli-Juuti

This paper describes a study that compared volatility distributions derived from direct evaporation measurements to estimates derived from measurements made using a FIGAERO-CIMS. Given the central role gas-particle partitioning plays in determining the amount of organic aerosol, this is an important topic. The FIGAERO-CIMS provides information on composition as a function of evaporation temperature but it has not been

widely evaluated in the context of figuring out volatility distribution of complex aerosols.

The paper describes a detailed analysis of previously published, relatively limited set of data – SOA formed from alpha-pinene formed in a oxidation flow reactor at two different O:C levels (low and medium). They consider low and high relative humidity and two different residence times. The very limited amount of the data is a real limitation to the paper.

Major comments/issues that broadly apply to the manuscript –

I found the paper to be a super detailed methods paper. It was not clear why it was submitted to ACP and not a journal like AMTD or AST. Based on the way the paper is written now, those journals are a better fit for the manuscript. My feeling is that while the topic of organic aerosol volatility is relevant to ACP this paper seemed is a very narrow and specialized for that journal (to me it read like a physical chemistry methods papers).

I had a hard time interpreting some the figures (e.g. Figure 2), which were often very detailed and contained many comparisons. For example, do you really need to show the three different VDevap results on Figures 2, 3 and 7 3 given they are basically the same – seems like an SI detail to help the focus the figure on what is important. This results in the paper have a bit of kitchen sink feel.

A closely related comment to the previous one, while the text described the figures and the results, I found it lacking in discussion of the results were telling us, specifically around this technique. There was too much focus on describing the data versus what the data are telling us about the technique and aerosol volatility. If it is to be accepted in ACP, I think the paper should be extensively rewritten to make it more accessible and understandable to the ACP audience. I spent a fair bit of time on the paper and got repeatedly bogged down. E.g. essentially all the tables can easily be moved to SI (maybe keep a very collapsed version of Table 1) because they are likely not of interest to a general ACP reader.

The very limited data set (a handful of conditions) seems like a pretty large limitation. Even for this limited set of data the method appears to not work so well for some conditions (e.g. low O:C in Figure 4b). There was also no discussion of experimental repeatability. There is probably enough data to justifying publication but this limitation of applying to a very narrow set of systems (and potentially overinterpreting the results) needs to be explicitly stated.

FIGAERO-CIMS – Given that it uses chemical ionization with iodide as the reagent ion as opposed to electron impact ionization, there are always concerns about mass closure. What fraction of the SOA mass is being detected by the instrument? If a large fraction of the aerosol mass that is not, then that seems like a big problem. This issue needs to be explicitly discussed, including its implications for measuring volatility. More molecular ions is a big advantage, but not measuring a large fraction of the material seems like a huge limitation as you are trying to draw inferences about the bulk aerosol based on characterizing only a limited fraction of the aerosol.

A closely related FIGAERO-CIMS concern – The FIGAERO only ramps to 200 oC (line 200). This is likely much too low to evaporate all of the SOA. Can the authors estimate what fraction of mass is evaporated? Do they account for that in any way? This issue needs to explicitly addressed in manuscript.

Uniqueness of fit – The paper takes a very empirical approach of fitting data to extract volatility distributions. That is fine and expected given the complexity of the aerosols. However, the number of data points is often quite small, comparable to the number of free parameters. For example, Figure 4 shows $\sim$7 data points that are fit to determine VDevap. How many free parameters are in the VDevap model? Line 516 indicates that you are fitting both C* and viscosity parameters – that is a lot of free parameters given the amount of data. The fit is clearly very underconstrained. The result is that there are likely many other solutions are close to the nominally best solution. This is an optimization problem and I suspect that the optimization function looks more like a plateau then a sharp peak therefore (within experimental uncertainty) there are likely

many good solutions. While I gave one example where this occurs, this is a general issue with the paper. For example, I am concerned about the same problem for the FIGAERO PMF approach, because that also have many free parameters. Presumably all of these solutions are reasonably close across the data, but I suspect with diverge as one extrapolates away from the data. This is a major issue with these sorts of empirical approaches. The paper needs to explicitly address this issue.

Here are some specific (but not exhaustive) comments (I spent several hours on this review but was unable to sort through all the details, even though I have published a fair bit on the topic of organic aerosol partitioning).

Line 65 – The paper highlights inconsistency between growth and evaporation experiments. Ultimately C* is a thermodynamic property (certainly at the molecular level) so these inconsistencies point to changing aerosol composition or other properties. Some of this sort of framing may be useful. I.e. if the volatility distribution of the aerosol is really changing, then presumably this reflects some other changes in composition that alters the underlying C* values. Alternatively there could be issues with the kinetics of evaporation. The authors are familiar with all these issues but the introduction might be improved with this framing. To me the issue seems more fundamental then measuring a volatility distribution.

Line 327 – "matches better" Based on what quantitative metric? This is one example of a broader issue of providing quantitative metrics of goodness of fit.

Table 1 – The analysis appears to have used an accommodation coefficient of 1 to interpret the evaporation data (alpha in Table 1 versus the alpha in equation (3)). This was not discussed or justified (none of the values were in Table 1 were justified). There are papers that report smaller values for this system (e.g. Saleh et al. Env. Sci. Tech. 2013). How would reducing this value alter the results from the analysis? This should be discussed in the paper.

Figure 2 indicates little agreement in the "raw" volatility distributions between the PMF

and evaporation approach. This is mentioned but not discussed in the text. The figure is also very confusing since there are overlaps in the volatility of the different PMF factors (i.e. one you don't show a volatility distribution of the PMF factors). I think it would be much clear if you lumped the factors together into a volatility distribution, using the colors to indicate the contribution of each factor to each bin (i.e. the volatility distribution would have stacked colored bars).

Compared to Figure 2, there is better agreement in Figure 3 when the offers have lumped in the material into larger bins wider than one order of magnitude. That seems encouraging, but this lumping process and its justification was not \discussed in the text. How different are the evapograms (Figure 4) when you use these different representations? A key issue is what level of information is there in the data. As I have discussed in earlier comment this problem seems very under constrained given the amount of data they have collected.

Line 109 – What compounds were used to calibrate Tmax? The paper should provide a calibration curve showing these results.

Figure 4 – It is interesting that the PMF approach performs better for the medium O:C aerosol compared to the low O:C aerosol. The PMF approach overestimates the evaporation of the This is mentioned on line 327, but only briefly discussed ($\sim$ line 500). The authors speculate it may be due to viscosity or particle phase chemistry. It would be good to think about whether this indicates a shortcoming of the approach. For example, could this be due to the CIMS not detected a larger fraction of the less oxygenated aerosol? (This is related to some earlier comments).

Figure 8. Optimized results (around line 440). The agreement seems impressive, but I suspect this is a just the result of fitting a model with lots of free parameters to a small set of data. Therefore, I am not surprised that the fit is great. If the fit is under constrained is this telling us anything about the technique? It was not clear if the authors had tested the applicability of the extracted values from this optimization

against data the model had not been fit to? If this optimization is to be presented these issues must be discussed and some sort of cross-validation presented.

Figure 8 –I interpret the dashed lines as the range of predications for the FIGAERO based approach (i.e. using the "theory" to predict viscosity). Is that correct? It is hard to tell what comparisons are based on truly independent comparison just using the FIGAERO versus fits of the data.

Figure 8. Can't differentiate between grey and black dashed lines.

[Figure]

---

## Author Comment (AC1) · 22 Apr 2020

We thank the reviewers for carefully reviewing our manuscript. Please see our answers to your comments in the attached PDF file.

Please also note the supplement to this comment:
https://www.atmos-chem-phys-discuss.net/acp-2019-927/acp-2019-927-AC1-supplement.pdf

---

## Author Response (AR1)

**Authors' response to comments received to manuscript** *"Comparing SOA volatility distributions derived from isothermal SOA particle evaporation data and FIGAERO-CIMS measurements"*

We thank both reviewer #1 and reviewer #2 for reviewing our manuscript and for the insightful comments that helped to improve the manuscript. Below we address the comments presented by the reviewers. The comments of the reviewers are **shown in bold,** our answers are shown as normal text, and the changes made to the manuscript are *shown in italic*. To improve readability we have numbered the comments of reviewer #2. All the line numbers given refer to the revised version of the manuscript.

**Please note** that the parametrization used for calculating effective saturation mass concentration $C^*$ from the desorption temperatures of each PMF factor is different in the revised version of the manuscript than in the ACPD version of the manuscript. The parameterization used in the ACPD version corresponded for a different type of FIGAERO and we have now revised the results using a parametrization applicable for the FIGAERO used in this study. The parametrization used in the revised version results in lower $C^*$ than the parametrization used in the ACPD version with the same desorption temperature input. This correction did not change our main conclusions. However, the new parametrization affects the results discussed in the comments 10 and 15 by the reviewer #2.

**Reviewer #1**

**In this manuscript, Tikkanen et al. compare SOA volatility derived from two different analyses: isothermal evaporation data and PMF applied to FIGAERO-CIMS data.Volatility is a critical property of organic aerosol, and I agree with the authors that the volatility data from FIGAERO-CIMS measurements have been under-utilized. This manuscript focuses on the comparison of the FIGAERO-CIMS PMF volatility data to isothermal evaporation data; the details of the FIGAERO-CIMS PMF volatility analysis are described in another manuscript currently under review (Buchholz et al., 2019 https://www.atmos-chem-phys-discuss.net/acp-2019-926/). It seems to me that this manuscript can only be accepted if Buchholz et al., 2019 is also accepted. I also request that my specific comments below be addressed before publication of this manuscript.**

The companion manuscript Buchholz et al. (2019b) is now at the "Editor Final Decision" status after some further minor revisions.

**1. The authors use data from previously conducted experiments in which SOA wasformed from alpha-pinene and ozone/OH. By varying experimental conditions, the SOA produced had low, medium or high O:C. Here, the authors only use experiments when the SOA formed had low or medium O:C. This choice is currently not explained or justified in the manuscript. In my opinion, the dataset utilized here is fairly limited,and the analysis would benefit from inclusion of these additional high O:C data. For**

**Example, overall the authors find that the agreement between the two volatility analysis better for intermediate O:C than for low O:C, and I am curious about the agreement at high O:C.**

Answer: The detailed analysis of the high O:C evapogram and particle composition data presented in Buchholz et al., (2019a and 2019b) strongly suggests an important influence of particle phase chemistry for these particles in the wet cases. For example, Buchholz et al., (2019a) show that one of the PMF factors has a significant contribution to the total thermogram in the wet RTC sample (sample taken at the later stages of the evaporation) even though the same factor is virtually non-existing in the fresh sample. This means that compounds were being formed during the isothermal evaporation experiment.

Including the high O:C results in our manuscript would require particle phase chemical reactions to be included in the model. Not enough is known about such reactions and therefore assumptions would need to be made about properties of the reaction products and the extent of the particle phase chemistry happening during the evaporation of the particles. This would lead to considerable uncertainty in the results. In our manuscript we wanted to keep the analysis as simple as possible and therefore elected not to include the more complex high O:C cases to the manuscript. Instead we included only the low and medium O:C cases for which Buchholz et al., (2019a,b) did not observe signs of significant effects from particle phase chemical reactions We have added the following explanation to line 91:

*The closer analysis of the high O:C experiments suggest particle phase reactions during the evaporation (Buchholz et al., 2019a,b). To avoid the uncertainty that would arise from unknown particle phase reactions, we chose not to include the high O:C data in our analysis.*

**2. The main takeaways from the manuscript should be clarified. The authors state in the abstract that "FIGAERO-CIMS measurements analyzed with the PMF method area promising method for inferring organic compounds' volatility distribution". The more detailed results point to the method working better under some conditions than others.It would be useful if the authors could make more concrete recommendations for future use of this method (PMF applied to FIGAERO-CIMS data) to obtain information on organic aerosol volatility.**

Answer: This comment is also linked to the first and third comment made by reviewer #2 about the discussion part of our manuscript. We have edited the discussion to better frame our findings. Please see our response to Reviewer 2.

**Editorial comments: There are several typographical and grammatical errors in the manuscript. I include a list of examples below:**

**Line 19: "volatility distributions derived the two ways are comparable within reasonable assumption"**

**Line 233: "and only evaporated th at different conditions"**

**Line 311: "To investigate the observed discrepancy more detailed"**

**Line 376: "In this section we compare VDPMF,opt of the fresh samples to VDPMF ofthe RTC sample to study are the two VD comparable."**

**Line 521-522: "thermogram data is good estimating the volatility distribution of organic aerosols"**

We thank reviewer #1 for pointing out these errors. We have corrected the ones presented here and also other typographical errors we found in the manuscript.

**Reviewer #2**

**Review of acp-2019-927 "Comparing SOA volatility distributions derived from isothermal SOA particle evaporation data and FIGAERO-CIMS measurements" by Olli-PekkaTikkanen, Angela Buchholz, Arttu Ylisirniö, Siegfried Schobesberger, Annele Virtanen,and Taina Yli-Juuti**

**This paper describes a study that compared volatility distributions derived from direct evaporation measurements to estimates derived from measurements made using a FIGAERO-CIMS. Given the central role gas-particle partitioning plays in determining the amount of organic aerosol, this is an important topic. The FIGAERO-CIMS provides information on composition as a function of evaporation temperature but it has not been widely evaluated in the context of figuring out volatility distribution of complex aerosols.The paper describes a detailed analysis of previously published, relatively limited set of data – SOA formed from alpha-pinene formed in a oxidation flow reactor at two differentO:C levels (low and medium). They consider low and high relative humidity and two different residence times. The very limited amount of the data is a real limitation to the paper.**

**1. Major comments/issues that broadly apply to the manuscript –I found the paper to be a super detailed methods paper. It was not clear why it was submitted to ACP and not a journal like AMTD or AST. Based on the way the paper is written now, those journals are a better fit for the manuscript. My feeling is that while the topic of organic aerosol volatility is relevant to ACP this paper seemed is a very narrow and specialized for that journal (to me it read like a physical chemistry methods papers).**

Answer: In our manuscript we show that volatility information derived from FIGAERO-CIMS data is in agreement with the volatility information derived from isothermal evaporation experiments. Considering the recent popularity of the FIGAERO-CIMS instrument in laboratory and atmospheric studies, we feel that this finding is of general interest to the ACP audience.

We agree that the presentation of our manuscript is somewhat more technical than the ACP audience may expect. We have edited the discussion part of our manuscript to better frame our findings and balance the text against the technical details of our study.

**2. I had a hard time interpreting some the figures (e.g. Figure 2), which were often very detailed and contained many comparisons. For example, do you really need to show the three different VDevap results on Figures 2, 3 and 7 3 given they are basically the same – seems like an SI detail to help the focus the figure on what is important. This results in the paper have a bit of kitchen sink feel.**

Answer: Thank you for this suggestion which helped as to clarify the presentation. We have edited the figures and tables the following way:
  - We moved the old figures 2, 3, 4, 5, 6 and 7 and tables 2 and 3, which show the $VD_{PMF}$ derived assuming different sample evaporation times to the supplementary material.
    - The new figures 2, 3, 4, 5, 6 and 7 and table 2 show the analysis only at the mean PMF sample evaporation time for the fresh samples (and at minimum PMF sample evaporation time of the RTC samples in figures 4 and 5 and table 2).
    - The old table 2 has been moved entirely to the supplement
    - The captions of the figures 2, 3, 4, 5, 6 and 7 have been edited to describe the new figures better

Due to these changes in the presentation of the figures and tables we have also edited the main text where the figures are described in the results section.

**3. A closely related comment to the previous one, while the text described the figures and the results, I found it lacking in discussion of the results were telling us, specifically around this technique. There was too much focus on describing the data versus what the data are telling us about the technique and aerosol volatility. If it is to**

**be accepted in ACP, I think the paper should be extensively rewritten to make it more accessible and understandable to the ACP audience.**

Answer: We have expanded the discussion and conclusion parts of the manuscript to make it more accessible to the general ACP audience and to highlight the main findings. The changes are on lines 505-507, 515-517, 519-523, 540-543, 548-550, 554-557, 562-563.

**4. I spent a fair bit of time on the paper and got repeatedly bogged down. E.g. essentially all the tables can easily be moved to SI(maybe keep a very collapsed version of Table 1) because they are likely not of interest to a general ACP reader.**

Answer: Please see our answer to comment 2

**5. The very limited data set (a handful of conditions) seems like a pretty large limitation.Even for this limited set of data the method appears to not work so well for some conditions (e.g. low O:C in Figure 4b). There was also no discussion of experimental repeatability. There is probably enough data to justifying publication but this limitation of applying to a very narrow set of systems (and potentially overinterpreting the results) needs to be explicitly stated.**

Answer: In figure 4 we compare the PMF VD with $C^*$ calculated from peak temperature value ($T_{max}$) of each factor to $VD_{evap}$. As noted in the results section, the $T_{max}$ value is not adequate for calculating $C^*$ when detailed particle dynamics (i.e. evaporation) is modelled. The $VD_{PMF,opt}$ whose $C^*$ values are optimized to match the evaporation data is able to capture the evaporation dynamics. The optimization fails only when we assume that the PMF sample represents the evaporating aerosol particles at the start of the fresh sample collection interval Overall the volatilities from FIGAERO-CIMS and isothermal evaporation agree for all our cases as long as the uncertainties in $C^*$ are taken into account.

We agree that the data sets available is a narrow one and we have added a note about this limitation in Section 5 lines 540-543

*We compared the two methods for obtaining the volatility distribution data for two different particle compositions and two evaporation conditions. The results are promising and suggest that the methods provide volatility distributions that are in agreement. We note that the data set available here is limited and additional investigations on comparing the methods are desirable in the future.*

The base case (low O:C a-pinene) has been studied in our lab in five separate measurement campaigns and the isothermal evaporation is the same within measurement error in all cases.

Also in a later campaign, which is not part of this study as the detailed design and settings of the FIGAERO-CIMS were different, we did repeat FIGAERO-CIMS measurements of the same type of aerosol on multiple days. The behavior of the identified PMF factors is the same between the different samples – only the contribution of background and contamination factors changed significantly as the circumstances changed on different days (e.g. switching to a new filter in FIGAERO). We therefore have a good reason to expect that the results presented here were repeatable even though repeated experiments were not included in this study.

**6. FIGAERO-CIMS – Given that it uses chemical ionization with iodide as the reagent ion as opposed to electron impact ionization, there are always concerns about mass closure. What fraction of the SOA mass is being detected by the instrument? If a large fraction of the aerosol mass that is not, then that seems like a big problem. This issue needs to be explicitly discussed, including its implications for measuring volatility. More Molecular ions is a big advantage, but not measuring a large fraction of the material seems like a huge limitation as you are trying to draw inferences about the bulk aerosol based on characterizing only a limited fraction of the aerosol.**

The $I^-$ anion in an iodide CIMS preferably clusters with molecules which contain hydroxyl-, hydroperoxyl-, carboxyl- or peroxycarboxyl- groups in their structure. Most products of the reaction of a-pinene with OH or $O_3$ contain two or more of these functional groups. Thus, the majority of them will be detectable with iodide CIMS even though it is more selective than EI. Mass closure studies for a-pinene SOA generated in a smog chamber have been conducted by Isaacman-VanWertz et al., (2017, 2018) comparing FIGAERO-CIMS to measurements with an SMPS (non-mass spectrometry technique) and a High-Resolution time of flight Aerosol Mass Spectrometer (AMS, Aerodyne research Inc., EI ionisation). They observed very good agreement for the detected particle phase carbon if FIGAERO-CIMS was calibrated as they laid out in the Supplement Material to Isaacman-VanWertz et al., (2018). The compounds produced in our study are comparable and thus a similarly good mass closure could be expected if similar sensitivity calibrations had been conducted for our FIGAERO system. However, such calibration is onerous and not available for the datasets at hand. So, by using uncalibrated FIGAERO-CIMS data here, we are implicitly assuming that the sensitivity towards individual compounds is uncorrelated to the compounds' volatility. We are not aware of published research against which to clearly test that assumption, but it appears plausible that less volatile compounds tend to be detected at higher sensitivity (Iyer et al., 2016; Lee et al., 2014). To our aid comes the maximum sensitivity (corresponding to ionization at the kinetic limit), which is obtained, e.g., for most di-carboxylic acids (#C>3). But it is likely that a bias is introduced that shifts FIGAERO-derived SOA compositions towards lower volatility. Indeed, if such a bias was accounted for, it could bring the evapograms modelled using $VD_{PMF}$ closer to the observations, as in particular the initial (fast) evaporation of relatively volatile material may be systematically underestimated when relying on (uncalibrated) FIGAERO data [Note that such bias would less clearly apply to observed desorption signals that are due to thermal decomposition, so, e.g., PMF factors associated with decomposition would still lead to high estimates when their $T_{max}$ is translated to C* (i.e. opposite bias, towards higher volatility), as discussed in Section 3.1.]

We have added the following to the revised version of the manuscript to lines 113-116 to clarify:

*Previous studies using FIGAERO-CIMS with iodide as the reagent ion found 50% or better mass closure compared to more established methods of quantifying OA mass (albeit with high uncertainties; (Isaacman-VanWertz et al., 2017; Lopez-Hilfiker et al., 2016). Therefore, it appears that the bulk of reaction products expected from a-pinene oxidation contains the functional groups required for detection by our FIGAERO-CIMS.*

And to lines 197-202:

*One more potential source of bias is our implicit assumption of a constant sensitivity of the CIMS towards all compounds, which follows from the lack of calibration measurements for our datasets (which indeed is a challenging endeavour; e.g., Isaacman-VanWertz et al., (2018)). It is plausible that less volatile compounds tend to be detected at higher sensitivity (Iyer et al., 2016; Lee et al., 2014), up to a kinetic limit sensitivity. Consequently, a volatility distribution derived from FIGAERO-CIMS thermograms may be biased towards lower volatility ($C^*$ bins), at least for compositions not associated with thermal decomposition.*

**7. A closely related FIGAERO-CIMS concern – The FIGAERO only ramps to 200 oC (line200). This is likely much too low to evaporate all of the SOA. Can the authors estimate what fraction of mass is evaporated? Do they account for that in any way? This issue needs to explicitly addressed in manuscript.**

Answer: As described in the reply to the previous comments, with appropriate sensitivity calibrations, the FIGAERO instrument ramping up to 200 °C gave mass closure with other instruments.

Our (as well as other's, e.g. Mohr et al., 2018) $C^*$ calibration suggests that at 200 °C the corresponding $C^*$ value is $\sim 10^{-14}$ µg m$^{-3}$, meaning compounds with this $C^*$ value will desorb at that temperature. This means that the recently defined class of ultra low volatility compounds (ULVOC, Schervish and Donahue, 2020) starting from $C^*$ values of $10^{-8.5}$ µg m$^{-3}$ would still be detected. Additionally, one has to keep in mind that many of these E/ULVOC, especially dimers, will decompose at temperatures below their theoretical desorption temperature. The thermal decomposition products will have a much higher volatility and are detected as such. Most of the compounds assigned to the D-type factors are products of such decomposition. This is discussed in section 3.1.

**8. Uniqueness of fit – The paper takes a very empirical approach of fitting data to extract volatility distributions. That is fine and expected given the complexity of the aerosols. However, the number of data points is often quite small, comparable to the**

**number of free parameters. For example, Figure 4 shows~7 data points that are fit to determine VDevap. How many free parameters are in the VDevap model?**

**Line 516 indicates that you are fitting both C* and viscosity parameters – that is a lot of free parameters given the amount of data. The fit is clearly very underconstrained. The result is that there are likely many other solutions are close to the nominally best solution. This is an optimization problem and I suspect that the optimization function looks more like a plateau then a sharp peak therefore (within experimental uncertainty) there are likely many good solutions.**

**While I gave one example where this occurs, this is a general issue with the paper. For example, I am concerned about the same problem for the FIGAERO PMF approach, because that also have many free parameters. Presumably all of these solutions are reasonably close across the data, but I suspect with diverge as one extrapolates away from the data. This is a major issue with these sorts of empirical approaches. The paper needs to explicitly address this issue.**

Answer: Here we address only the optimization done to high RH data. The comment refers also to low RH data which we will address in comment 16 where the optimization to low RH data is brought up again.

In an optimization run the amount of data point is small compared to the number of free parameters. When VD mole fractions are optimized based on only the evapogram data ($VD_{evap}$) the number of free parameters is equal to the number of VD bins minus 1 as the mole fractions must sum to one.  When the $C^*$ values are estimated from the high RH data the number of free parameters is equal to the number of PMF factors. However, in the latter case we are not optimizing a completely unconstrained model to the measurements. The C* values are given constraints from the PMF analysis and our goal is to inspect if it is possible to explain the observed evaporation with these values.

Figure AR1  show how the estimated $C^*$ values are distributed among the 50 independent optimization runs performed for each fresh sample and mean sample evaporation time. The figure show that when the $C^*$ of a factor affects the evaporation dynamics i.e. the minimum and/or maximum value of a factor is inside the red dashed lines the $C^*$ values do not change much between different optimization runs. Note that the spread of values can become wider for a factor when its contribution to the total signal is close to zero (e.g. factor MD1a Fig AR1a or LD1a in Fig AR1b).

[Figure]

Figure AR1: Box plots showing how the estimated $C^*$ of PMF factors are distributed in 50 independent optimization runs of high RH fresh samples. a) medium O:C mean sample evaporation time b) low O:C mean sample evaporation time Black circles show the minimum and maximum possible value allowed in the optimization (based on the thermograms of the PMF factors) and red dashed lines show the minimum and maximum $C^*$ value that can be estimated from the isothermal evaporation measurements.

**9. Here are some specific (but not exhaustive) comments (I spent several hours on this review but was unable to sort through all the details, even though I have published a fair bit on the topic of organic aerosol partitioning).**

**Line 65 – The paper highlights inconsistency between growth and evaporation experiments. Ultimately C\* is a thermodynamic property (certainly at the molecular level) so these inconsistencies point to changing aerosol composition or other properties. Some of this sort of framing may be useful. I.e. if the volatility distribution of the aerosol is really changing, then presumably this reflects some other changes in composition that alters the underlying C\* values. Alternatively there could be issues with the kinetics of evaporation. The authors are familiar with all these issues but the introduction might be improved with this framing. To me the issue seems more fundamental then measuring a volatility distribution.**

Answer: In line 65 we brought up the difference between SOA evaporation and growth measurements to point out that the volatility information derived from experiments depends on the experiment setup. The limitations of the methods raise a need to develop new tools for extracting the volatility information of SOA constituents.

We have modified the text in line 68-73 in the revised version of the manuscript to

*However, the experimental setup also defines the range of $C^*$ values that can be estimated from the data. Vaden et al., (2011) and Yli-Juuti et al., (2017) have both shown that the volatility basis sets derived from SOA growth experiments results in too fast SOA evaporation compared to measured evaporation rates when used as input for process models. Possible reasons for such discrepancies include the different $C^*$ ranges to which the SOA growth and SOA evaporation experiments are sensitive and the role of vapor wall losses in SOA growth experiments.*

**10. Line 327 – "matches better" Based on what quantitative metric? This is one example of a broader issue of providing quantitative metrics of goodness of fit.**

Answer: Note that the simulated evapogram curves in Fig. 4 have changed as we corrected the error in the $T_{max}$-$C^*$ calibration. The evapograms calculated with the $VD_{PMF}$ of the medium O:C RTC sample produce almost equal evapogram as the one calculated with $VD_{evap}$. We have adjusted the text accordingly. Also, we increased the readability of the figure by showing only the simulations with medium evaporation time for the FIGAERO samples.

The reviewer is correct, that this is a purely qualitative term and whenever suitable one should use objective and mathematically based parameters for such comparisons. But for this specific example we decided to use a qualitative description rather than goodness of fit statistics. We are comparing the overall shape of the simulated and measured evapograms, but for the evapogram curves simulated with RTC $VD_{PMF}$ we have only 1 or 2 measurement points to directly compare to. It is very clear that the evapograms simulated with the fresh $VD_{PMF}$ underestimate the evaporation (too slow evaporation) for medium O:C particles while those using the RTC $VD_{PMF}$ create curves that display a very similar shape as those simulated with $VD_{evap}$ and estimated from the measured points. In the low O:C case, we now see a slight underestimation of evaporation rate using the fresh $VD_{PMF}$ and overestimation of evaporation rate with the RTC $VD_{PMF}$. A simple goodness of fit parameter like the mean squared error would not reflect the direction of this discrepancy as a qualitative description can. Looking at the revised Fig. 4a where we show only one brown line we feel that "matches well" describes well what we want to say.

**11. Table 1 – The analysis appears to have used an accommodation coefficient of 1 to interpret the evaporation data (alpha in Table 1 versus the alpha in equation (3)). This was not discussed or justified (none of the values were in Table 1 were justified). There are papers that report smaller values for this system (e.g. Saleh et al. Env. Sci.**

**Tech.2013). How would reducing this value alter the results from the analysis? This should be discussed in the paper.**

Answer: The theoretical framework used by Saleh et al., (2013) assumes that the mass-accommodation coefficient includes any mass transfer limitations caused by high viscosity of the particle phase. In Saleh et al., (2013) SOA growth and evaporation experiments were performed with α-pinene ozonolysis SOA at 10% RH. Based on the work of Li et al., (2019) it is likely that there are significant mass transfer limitations associated with this type of SOA and RH. Those mass transfer limitations likely decrease the mass-accommodation coefficient in Saleh et al., (2013).

In our work, we model the mass transfer limitations explicitly with the KM-GAP model and thus the mass-accommodation coefficient in our work consists of effects due to e.g. surface sticking which we neglect. Additionally, the work of Julin et al., (2014) reports near-unity mass-accommodation coefficients for various organic molecules based on molecular dynamic simulations and experiments.

We have added justification for the properties of the organic compounds presented in Table 1 as a footnote to the table

*[b] values are chosen to represent a generic organic compound with values similar to other α-pinene SOA studies (e.g. Pathak et al., 2007; Vaden et al., 2011; Yli-Juuti et al., 2017).*

**12. Figure 2 indicates little agreement in the "raw" volatility distributions between the PMF and evaporation approach. This is mentioned but not discussed in the text. The figure is also very confusing since there are overlaps in the volatility of the different PMFfactors (i.e. one you don't show a volatility distribution of the PMF factors). I think it would be much clear if you lumped the factors together into a volatility distribution,using the colors to indicate the contribution of each factor to each bin (i.e. the volatility distribution would have stacked colored bars). Compared to Figure 2, there is better agreement in Figure 3 when the offers have lumped in the material into larger bins wider than one order of magnitude. That seems encouraging, but this lumping process and its justification was not\discussed in the text.**

Answer: In Figure 2 we show the information from PMF analysis that we utilized in the model simulations. Therefore, the range of C* values for each factor is an important piece of information and we find that combining the factors in stacked bars would not convey this information as directly. Our motivation for the lumping process was that when we examine the "raw" volatility distribution of Figure 2, it is not evident if the two VD are similar as the reviewer also points out. One issue complicating the comparison is that the PMF analysis does not set uni-distant $C^*$ values for the factors. The second point is that the FIGAERO samples can differentiate compounds with $C^*$ values below -2. These compounds do not evaporate under the investigated isothermal evaporation conditions and are thus already grouped into the

lowest volatility bin in VD$_{evap}$. We lumped both volatility distributions to volatility classes to study the similarities between the distributions on a qualitative level. The justification for the choice of these volatility classes is given in lines 325-328 of the manuscript. Finally, as a quantitative comparison we study what kind of evapograms the "raw" (i.e. "non-lumped") VD would produce. We have clarified our reasoning in the beginning of the results section on line 280-283:

*We investigate the VD both on a qualitative and quantitative level. On a qualitative level we compare the amount of matter of different C\* intervals. On a quantitative level we study what is the evaporation behavior of the particles based on the determined VD and how they compare to the measured evaporation.*

**13. How different are the evapograms (Figure 4) when you use these different representations? A key issue is what level of information is there in the data. As I have discussed in earlier comment this problem seems very under constrained given the amount of data they have collected.**

Answer: We do not use the three lumped volatility classes VD in the model simulations. The evapograms are always calculated with the "raw" VD (the one shown in Fig. 2). We hope that the modifications of the text mentioned above in comment 12 clarifies this.

**14. Line 109 – What compounds were used to calibrate Tmax? The paper should provide a calibration curve showing these results.**

Answer: We used polyethylene glycols (PEG) solutions in acetonitrile with 5 to 8 glycol units. As requested, we added the calibration curve and a brief description to the SI material.

**15. Figure 4 – It is interesting that the PMF approach performs better for the mediumO:C aerosol compared to the low O:C aerosol. The PMF approach overestimates the evaporation of the This is mentioned on line 327, but only briefly discussed (~line500). The authors speculate it may be due to viscosity or particle phase chemistry. It Would be good to think about whether this indicates a shortcoming of the approach. For example, could this be due to the CIMS not detected a larger fraction of the less oxygenated aerosol? (This is related to some earlier comments).**

Answer:  The Figure 4 has changed in the revised version of the manuscript as we corrected the error in the T$_{max}$-C\* calibration. With the new parametrization it looks like the PMF approach (VD$_{PMF}$) performs better for the low O:C aerosol compared to medium O:C aerosol. In both

oxidation conditions the evapogram calculated with $VD_{PMF}$ of the fresh samples shows less evaporation than the measurements or the evapograms calculated with $VD_{evap}$.

The point raised by the reviewer is a valid one. The PMF method lumps all the organic compounds detected by the CIMS into preset number of factors. These factors are then treated as surrogate compounds when the evapograms are calculated with the LLEVAP model in Fig. 4. Given that in $VD_{PMF}$ only one value is assigned to $C^*$ of every PMF factor, it is not surprising that the $VD_{PMF}$ does not produce an evapogram similar to the measurements.

Because the $VD_{PMF}$ underestimates the evaporation, it seems possible that the iodide CIMS does not detect some fraction of the less oxygenated organic compounds, as we mentioned in our answer to comment 6. In our work, we expect that majority of the compounds are detected in the CIMS and the disagreement between measured and simulated evapograms in Fig. 4 comes from lumping the organic compounds into surrogate compounds in the PMF analysis or from the uncertainty in the conversion of desorption temperature to $C^*$.

**16. Figure 8. Optimized results (around line 440). The agreement seems impressive, but I suspect this is a just the result of fitting a model with lots of free parameters to a small set of data. Therefore, I am not surprised that the fit is great. If the fit is under constrained is this telling us anything about the technique? It was not clear if the authors had tested the applicability of the extracted values from this optimization against data the model had not been fit to? If this optimization is to be presented these issues must be discussed and some sort of cross-validation presented.**

Answer: We agree that in the case of optimizing simultaneously $C^*$ and viscosity parameters there are a lot of free parameters (although neither $C^*$ or $b_i$ are completely free parameters as they are restricted with minimum and maximum values) and therefore the estimates of parameter values from such optimizations should be interpreted with caution. However, our motivation with the low RH case was not to derive a universal parameterization. Our interest with the low RH data was to perform a cross-validation using a parametrization developed previously based on measurements of glass transition temperature of various organic compounds (DeRieux et al., 2018). Such validations of parameterizations against SOA dynamics are of importance if the parameterizations are to be used in the future e.g. for interpreting ambient or laboratory measurements or in large-scale model simulations of SOA formation. Our results show that the viscosity of SOA can be captured with this parametrization given the uncertainty in the parametrization and the $C^*$ values that we estimated using the same approach as with the high RH data. We have clarified our intent with the low RH experiments by adding following to lines 426-430

*Our aim is to test if the slower evaporation, presumably due to higher viscosity of the SOA can be captured with a recently developed viscosity parametrization based on glass transition temperatures of various organic compounds (DeRieux et al., 2018). We also compare the results using the viscosity parametrization to an approach where we fit both the viscosity and VD to the evapogram.*

**17. Figure 8 –I interpret the dashed lines as the range of predications for the FIGAERO based approach (i.e., using the "theory" to predict viscosity). Is that correct? It is hard to tell what comparisons are based on truly independent comparison just using the FIGAERO versus fits of the data.**

Answer: The reviewer is correct. We have added following clarification to the caption of Fig. 8

*Grey lines show the minimum and maximum possible evaporation calculated with $VD_{PMF,dry}$ ($C^*$ of PMF factors calculated from $T_{max}$) and the highest (the original parametrization of DeRieux et al., (2018), grey dashed lines) or the lowest (30 K substracted from the $T_g$ of every ion, grey solid line) studied viscosity.*

**18. Figure 8. Can't differentiate between grey and black dashed lines.**

Answer: We have changed the color of both lines to grey and changed one line to be solid and the other dashed.

**References**

Buchholz, A., Ylisirniö, A., Huang, W., Mohr, C., Canagaratna, M., Worsnop, D. R., Schobesberger, S. and Virtanen, A.: Deconvolution of FIGAERO-CIMS thermal desorption profiles using positive matrix factorisation to identify chemical and physical processes during particle evaporation, Atmos. Chem. Phys. Discuss., doi:https://doi.org/10.5194/acp-2019-926, 2019a.

Buchholz, A., Lambe, A. T., Ylisirniö, A., Li, Z., Tikkanen, O.-P., Faiola, C., Kari, E., Hao, L., Luoma, O., Huang, W., Mohr, C., Worsnop, D. R., Nizkorodov, S. A., Yli-Juuti, T., Schobesberger, S. and Virtanen, A.: Insights into the O:C-dependent mechanisms controlling the evaporation of α-pinene secondary organic aerosol particles, Atmos. Chem. Phys., 19(6), 4061–4073, doi:10.5194/acp-19-4061-2019, 2019b.

DeRieux, W.-S. W., Li, Y., Lin, P., Laskin, J., Laskin, A., Bertram, A. K., Nizkorodov, S. A. and Shiraiwa, M.: Predicting the glass transition temperature and viscosity of secondary organic material using molecular composition, Atmos. Chem. Phys., 18(9), 6331–6351, doi:https://doi.org/10.5194/acp-18-6331-2018, 2018.

Isaacman-VanWertz, G., Massoli, P., E. O'Brien, R., B. Nowak, J., R. Canagaratna, M., T. Jayne, J., R. Worsnop, D., Su, L., A. Knopf, D., K. Misztal, P., Arata, C., H. Goldstein, A. and H. Kroll, J.: Using advanced mass spectrometry techniques to fully characterize atmospheric organic carbon:

current capabilities and remaining gaps, Faraday Discussions, 200(0), 579–598, doi:10.1039/C7FD00021A, 2017.

Isaacman-VanWertz, G., Massoli, P., O'Brien, R., Lim, C., Franklin, J. P., Moss, J. A., Hunter, J. F., Nowak, J. B., Canagaratna, M. R., Misztal, P. K., Arata, C., Roscioli, J. R., Herndon, S. T., Onasch, T. B., Lambe, A. T., Jayne, J. T., Su, L., Knopf, D. A., Goldstein, A. H., Worsnop, D. R. and Kroll, J. H.: Chemical evolution of atmospheric organic carbon over multiple generations of oxidation, Nature Chem, 10(4), 462–468, doi:10.1038/s41557-018-0002-2, 2018.

Iyer, S., Lopez-Hilfiker, F., Lee, B. H., Thornton, J. A. and Kurtén, T.: Modeling the Detection of Organic and Inorganic Compounds Using Iodide-Based Chemical Ionization, J. Phys. Chem. A, 120(4), 576–587, doi:10.1021/acs.jpca.5b09837, 2016.

Julin, J., Winkler, P. M., Donahue, N. M., Wagner, P. E. and Riipinen, I.: Near-Unity Mass Accommodation Coefficient of Organic Molecules of Varying Structure, Environ. Sci. Technol., 48(20), 12083–12089, doi:10.1021/es501816h, 2014.

Lee, B. H., Lopez-Hilfiker, F. D., Mohr, C., Kurtén, T., Worsnop, D. R. and Thornton, J. A.: An Iodide-Adduct High-Resolution Time-of-Flight Chemical-Ionization Mass Spectrometer: Application to Atmospheric Inorganic and Organic Compounds, Environ. Sci. Technol., 48(11), 6309–6317, doi:10.1021/es500362a, 2014.

Li, Z., Tikkanen, O.-P., Buchholz, A., Hao, L., Kari, E., Yli-Juuti, T. and Virtanen, A.: Effect of Decreased Temperature on the Evaporation of α-Pinene Secondary Organic Aerosol Particles, ACS Earth Space Chem., 3(12), 2775–2785, doi:10.1021/acsearthspacechem.9b00240, 2019.

Lopez-Hilfiker, F. D., Mohr, C., D'Ambro, E. L., Lutz, A., Riedel, T. P., Gaston, C. J., Iyer, S., Zhang, Z., Gold, A., Surratt, J. D., Lee, B. H., Kurten, T., Hu, W. W., Jimenez, J., Hallquist, M. and Thornton, J. A.: Molecular Composition and Volatility of Organic Aerosol in the Southeastern U.S.: Implications for IEPOX Derived SOA, Environ. Sci. Technol., 50(5), 2200–2209, doi:10.1021/acs.est.5b04769, 2016.

Mohr, C., Lopez-Hilfiker, F. D., Yli-Juuti, T., Heitto, A., Lutz, A., Hallquist, M., D'Ambro, E. L., Rissanen, M. P., Hao, L., Schobesberger, S., Kulmala, M., Mauldin III, R. L., Makkonen, U., Sipilä, M., Petäjä, T. and Thornton, J. A.: Ambient observations of dimers from terpene oxidation in the gas phase: Implications for new particle formation and growth, Geophysical Research Letters, 2958–2966, doi:10.1002/2017GL072718@10.1002/(ISSN)1944-8007.GRLHIGHLIGHTS2017, 2018.

Pathak, R. K., Presto, A. A., Lane, T. E., Stanier, C. O., Donahue, N. M. and Pandis, S. N.: Ozonolysis of α-pinene: parameterization of secondary organic aerosol mass fraction, Atmos. Chem. Phys., 7(14), 3811–3821, doi:10.5194/acp-7-3811-2007, 2007.

Saleh, R., Donahue, N. M. and Robinson, A. L.: Time Scales for Gas-Particle Partitioning Equilibration of Secondary Organic Aerosol Formed from Alpha-Pinene Ozonolysis, Environ. Sci. Technol., 47(11), 5588–5594, doi:10.1021/es400078d, 2013.

Schervish, M. and Donahue, N. M.: Peroxy radical chemistry and the volatility basis set, Atmos. Chem. Phys., 20(2), 1183–1199, doi:10.5194/acp-20-1183-2020, 2020.

Vaden, T. D., Imre, D., Beránek, J., Shrivastava, M. and Zelenyuk, A.: Evaporation kinetics and phase of laboratory and ambient secondary organic aerosol., Proc. Natl. Acad. Sci. U.S.A., 108(6), 2190–2195, doi:10.1073/pnas.1013391108, 2011.

Yli-Juuti, T., Pajunoja, A., Tikkanen, O.-P., Buchholz, A., Faiola, C., Väisänen, O., Hao, L., Kari, E., Peräkylä, O., Garmash, O., Shiraiwa, M., Ehn, M., Lehtinen, K. and Virtanen, A.: Factors controlling the evaporation of secondary organic aerosol from α-pinene ozonolysis, Geophys. Res. Lett., 44(5), 2016GL072364, doi:10.1002/2016GL072364, 2017.

---

## Author Response (AR3)

Authors' response to comments received to manuscript "Comparing SOA volatility distributions derived from isothermal SOA particle evaporation data and FIGAERO-CIMS measurements"

We thank both reviewer #3 and reviewer #4 for reviewing our manuscript and for the helpful comments that helped to improve the manuscript. Below we address the comments presented by the reviewers. The comments of the reviewers are **shown in bold**, our answers are shown as normal text, and the changes made to the manuscript are *shown in italic*. We have numbered the specific comments of reviewer #4.

**Reviewer #3**

The authors have adequately addressed my concerns from the first iteration. I noticed some typographical errors in the added *I* revised text; hence the request for technical corrections.

We thank reviewer #3 for his/her comments. We have carefully read through the manuscript and corrected any typos found.

**Reviewer** #4**

The authors present here an investigation of how FIGAERO-CIMS thermal desorption data can be used to understand volatility distributions and evaporation kinetics, specifically through comparison to isothermal evaporation experiments. Overall, I think this is useful work of interest to the ACP community, and I think the authors have explored and considered in detail many of the potential areas of uncertainty in the approach. My opinion is that is generally suitable for publication in this journal. I have two main comments below that I think need to be discussed and addressed prior to publication, and list a number of specific comments below (in some cases, specific comments may just be specific examples/cases of the general comments and can be responded to as such).

Major comments:

1) Presentation. The authors are very detailed, but I fear this may contributed to the fact that I find this paper and these figures hard to get through. With all the different comparisons and model paramters, it takes a lot of re-reading sentences and mentally refreshing myself on the modeling frameworks to keep on top of what I am looking at. This is particularly true in considering all the variables in the evapogram modeling. Unfortunately I'm not sure I have a lot of concrete suggestions for how to fix this issue, it may just stem out of all the modeling details. I think one clear thing that might help would be to make the "takehome points" of the figures more clear within the figures themselves instead of relying entirely on captions and legends, for example: in all figures, including in each subplot what data set (e.g. "Fresh, medium O:C SOA") is being shown, in Figure 2 move most of the legend into the plot by just labeling "VD\_evap" in gray and "Factors" in colored text, etc. Other possible ways to increase clarity might be: refer to and label the volatility classes as "low", "semi-volatile", and "volatile" instead of 1, 2, and 3 so the read doesn't have to keep track; spend less time discussing (and labeling in figures) the issues around min, mean, and max evaporation

times since it doesn't really impact the conclusions and move most of that to the SI. A minor but still important issue is awkward phrasing and grammar - on top of general complexity of the presentation, there are a lots of grammatically questionable phrases and/or typos that should be fixed (some, but not all, are listed in the specific comments).

We thank reviewer #4 for these suggestions that helped to further clarify the presentation. We have made following changes to the figures in the main text:

- added the name of the data set to each subplot in Figures 1 8 in the manuscript
- added labels for each PMF factor in Figure 2 to make it easier to recognize the factors without looking the legends.
- Modified the legend texts in Figures 4 6 so that they are better in line with the main text and the captions
- Added text arrows to Figure 6 that explain what bar shows what data.

We have removed the mentions about min, mean and max evaporation times in lines 395, 401-402, 404-406, 424, 429-432, 436-439, 539-541 and in the figure captions in the highlighted version of the manuscript. Instead, we briefly describe the results concerning different evaporation times in the captions of the supplementary Figure S8

The simulations of the fresh samples that start at the mean or maximum evaporation time resemble the measured evaporation and the evaporation simulations calculated with the  $VD_{evap}$ . The simulation of the fresh sample that starts from the minimun evaporation time does not produce evaporation curve similar to the measurements. The results highlight the fact that it is not likely that the fresh sample consists of particles that have just entered the residence time chamber.

**and Figure S9**

The results of medium O:C SOA in high RH experiments show that the  $VD_{PMF}$  best resembles the  $VD_{PMF,opt}$  of the maximum evaporation time, although the difference to the mean evaporation time is not significant. For low O:C SOA in high RH experiments, the results show that the  $VD_{PMF}$  best resembles the  $VD_{PMF,opt}$  of the mean evaporation time.

We have corrected the typos and phrases pointed out in the specific comments as well as read through the manuscript carefully and corrected any typos / unclear sentences we found.

In principle we agree that it would be useful to label volatility classes 1,2 and 3 into "low", "semi-volatile" and "volatile" classes. In theory the volatility of a compound of course depends on the conditions where the volatility is studied and for our system it would be justified to name the three classes as "low-volatile", "semi-volatile" and "volatile". In practice we think that the terms "low-volatile", "semi-volatile" have settled within the SOA community to mean specific C\* ranges which are different from the C\* ranges than what we use in our study for our volatility classes 1, 2 and 3. To avoid confusion, we label the volatility classes with numbers, even though it is not the most convenient labeling.

2) Benefit of PMF. A core component of this work is the PMF analysis of the FIGAERO data, and the authors do a detailed investigation to understand thermogram-derived volatility. A major conclusion of this work is that PMF factors can seemingly be used to describe the volatility/evaporation, but only if you optimize the T --> C\* conversion by

fitting to evapogram data (i.e., account for the "uncertainty in the desorption temperature"). However, it's totally unclear to me that the PMF step is at all necessary, given the scale of that uncertainty, and the typical absence of evapogram data to provide that constraint. The C\* that describes each factor is uncertain to around an order of magnitude, bounded by the desorption temperature range of the factor. Given this range of uncertainty (which is typical for C\* estimates), what do you gain by having some specific T max or T range associated with each factor? Why not just cut the thermogram into VBS bins based on temperatures (either by cutoff temperatures, or by fitting peaks constrained to the temperatures defining each bin)? Would this approach do any worse a job in comparison to the VD evap or evaporation kinetics? This might even do better - M4, for instance, is very broad with T max near T 25, so forcing all mass in the factor to this range might drive some bias. There are reasons beyond volatility you might want PMF, but this paper does not convince me that the effort of PMF provides any benefit in estimating volatility, and if not, why is it being used at all? Something needs to be added to this manuscript to provide support or context for this decision, e.g., a discussion of other peoples use of PMF for volatility; a discussion of why PMF might reasonably be expected to do a better job than a simple VBS approach; or a comparison of the present PMF approach to a VBS-only approach. Ideally, I'd love to see the VBS-only approach applied because if it works it informs how one can use the thermogram without the more complex need for PMF, but I would understand if the editor and/or authors feel that is beyond the scope. If it is beyond the scope, I do think it should be considered and discussed as a possibility - one major takehome for me is that it seems like it should work at least within similar unceratinty, so I think it would broaden the audience and potential impact of this paper to explore the possibility.

There can be two important issues with deriving VBS distributions directly from the sum thermograms:

 There can be a considerable contribution of background and/or contamination distorting the sum thermogram. Especially, when the collected mass loading on the FIGAERO filter is low (few 10s of ng), it is important to separate the background from the real sample signal.

As also described in our reply to specific comment 21, the low O:C, fresh, dry sample had a significant contamination. In Figure AR1, we depict the measured (black circles) and reconstructed sum thermogram (blue line). The red line shows the reconstructed sum thermogram using only the contribution of L1-L5 and LD1, omitting the background and contamination. It is clear that "binning" these two sum thermograms will lead to different VBS distributions.

The PMF approach identifies the instrument background (and contamination) and allowed us to omit it from further analysis.

*Figure AR1: Measured and reconstructed sum thermogram for the low O:C, fresh, dry sample.*

2) The  $T_{max} \rightarrow C^*$  conversion is derived for single compounds relating their  $T_{max}$  values to their volatility. The area or the shape of the thermogram is not considered. Without further studies, it is not clear if this discrete calibration can be applied to the continuum of the sum thermogram.

A compromise could be to determine the  $T_{max}$  value for each single ion thermogram and then use this as the volatility measure. However, there are multiple ions with multimodal or broad single ion thermograms in our data set. A single  $T_{max}$  value per ion again ignores the shape of the single ion thermogram, and thus possibly overestimates the volatility.

The PMF analysis can be understood as "binning" the sum thermogram by using the information from the desorption of all ions. The calculation time is clearly longer than for a simpler approach. But the amount of information and the solution of the issues mentioned above more than justifies that.

As the reviewer states, there are also reasons beyond volatility why one might want to use the PMF method. For example, one such reason might be understanding the particle phase chemistry during SOA evolution as was done in our group in Buchholz et al., (2019).

Because the PMF method might be important for interpreting the properties of the SOA constituents on a different level than calculating physicochemical properties from the mass spectrometer data, we feel that it is important to study that the method can also represent the volatility, which is perhaps the most important property for SOA formation and evolution, correctly.

Comparing the PMF method to VBS-only approach is an intriguing idea, but we feel that it is out of the scope of this paper. We have added the following to the discussion part (lines 507-512) in the revised version of the manuscript about the different methods to estimate the volatility

In addition to the PMF method used here, also other ways of characterizing SOA compound volatilities or VBS from FIGAERO-CIMS thermograms have been suggested (e.g. Stark et al., 2017). These include, for example, the more straightforward method of calculating the C\* of each detected ion based on their  $T_{max}$ , using Eq. (3) and lumping them into a traditional VBS. While such other methods may capture the volatility distributions sufficiently, the benefit of PMF method is that it offers a new way to understand what happens inside the particles, e.g.

during the heating in FIGAERO. Here we have evaluated this method with respect to its ability to capture the volatilities of SOA.

**Specific comments:**

**1.** Line 40-41. This whole sentence is awkward English and difficult to understand, rephrase**

We have rephrased the sentence to:

There exist gaps in the knowledge especially on formation and deposition of SOA as well as how the processes are affected by changing physicochemical properties such as volatility (Glasius and Goldstein, 2016).

**2. Line 41. Should be phrased "the phase state...has also..."**

We have moved the word 'also' as suggested.

**3. Line 45. Mass Spectrometer should be capitalized**

We have capitalized Mass Spectrometer.

**4. Line 77. "conduction a" is a typo**

We have changed "conduction" to "conducting"

**5. Line 82. For "How to interpret..." is not a grammatically correct question, should be "How shoud....be interpreted?"**

We have modified the sentence as:

**How should the PMF results of FIGAERO-CIMS data be interpreted in terms of volatility?**

**6. Line 104-107. Do I understand correctly that these 80 nm particles sit in the 100 L chamber for 4-10 hours? Even a monodisperse population will have some size distribution - how do you account for particle-dependent wall losses that might change the apparent size distribution? These might be negligible, but if so, the authors should provide evidence to support such an assumption.**

The selected monodisperse distribution is narrow. We fitted the measured size distributions with an asymmetric log-normal function from which we get the half-width of the distributions and the peak position. For the low O:C, wet experiment (which exhibited the strongest evaporation) the half-width was 9-10 nm for the 75 – 80 nm particles at the start of the experiment. After 10 h, the particle size may decrease to as low as 55 nm with a half-width of 6-7nm. We do not expect significant size dependence of the particle wall loss for such narrow size distributions in the range of 80 - 50 nm.

**7. Line 111. I don't think you need to include Aerodyne Research Inc. here, you already reference it in the paranthetical at the end of the sentence.**

In line 111 we reference the Aerodyne Research Inc. first for the FIGAERO and later for the CIMS in the parenthesis. There are custom-build FIGAERO units thus it is necessary to indicate that the used unit was the model from Aerodyne Research Inc.

8. Line 114. It might be worth noting that in Isaacman-VanWertz et al. cited here, the volatility distribution of the FIGAERO-CIMS based on thermograms was similar to that of a TD-AMS. This suggests that any overall conclusions from the present work likely extend to other thermal desorption based estimates of volatility.

We would at least hope so. On the other hand, in Isaacman-VanWertz the authors test only one SOA system ( $\alpha$ -pinene oxidized by OH). In general it might be that depending on which system is being investigated, different experimental methods may perform better or worse in producing accurate volatility distributions. For this reason, we feel that it is perhaps better not to draw conclusions from the present work to other thermal desorption based estimates of volatility.

**9. Line 115. "a-" instead of "alpha-"**

We have changed "a-" to " $\alpha$ " in line 115 in the old version of the manuscript.

10. Line 143. Why not just call them "VD bins" in all cases, and avoid the confusion with "compounds"? Having read the paper in detail, it's not clear to me that the use of the term VD compound is at all necessary - each bin has average properties (e.g., T\_max) and I don't see any need to refer to them explicitly as psuedo-individual compounds.

Thank you for this suggestion. We have changed all occurrences of "VD compound" to "VD bin".

11. Line 144. Are these really the properties of each VD compound? It looks like just basic assumptions about the properties of all (not each) bin. This sentence makes me think the Table is going to contain a list of many different properties for each of the bins.

To avoid confusion we have changed the sentence in line 144 of the old version of the manuscript from

The physicochemical properties of each VD bin are listed in Table 1 as well as the ambient conditions of each evaporation experiment.

to

The physicochemical properties of each VD bin are assumed to be the same. These properties and the ambient conditions of each evaporation experiment are listed in Table 1.

**12.** Table 1. Use column and row dividers to make clear that the rows with only one value are for all columns**

We have added column dividers to Table 1.

**13. Line 219. "and conducting a" should read "and conduct a"**

We have changed this as suggested.

14. Line 223-224. I don't understand the purpose of this interpolation. Isn't the mass loading profile basically a representaion of mass as a function of time/temperature? What do the authors mean a temperature "step". Usually, the CIMS collects data at ~1 Hz - how large a step in temperature occurs in one second? Do the authors mean they

interpolate 100 spectra per degree C? Or 100 spectra per Hz? I would guess that if you are just interpolating from your existing data, this would not actually increase your statistical power, since the amount of "real" information is not increasing, but I'm not a statistician so I'm not sure.

The raw CIMS data had indeed a time resolution of 1 Hz. But because the raw signals of many ions were so low (and noisy), it was necessary to average the raw signals over a longer time period before the high-resolution analysis could be conducted. We used 20 s as averaging intervals which leads to an average  $\Delta T$  between two adjacent data points of ~4°C during the linear phase of the heating ramp.

While the overall shape of the thermograms is still visible, this T grid is to coarse to determine  $T_{max}$  and the  $25^{th}/75^{th}$  percentiles directly. One option would be to fit the thermograms with a function, assuming a peak shape (e.g. gaussian or log-normal). Instead we chose to apply a linear interpolation assuming a continuous, linear distribution between two adjacent data points.

We have modified the paragraph in lines 225-229 in the revised version of the manuscript to:

Once the PMF algorithm was applied to the FIGAERO-CIMS data we calculated the VD from the mass loading matrix **G**. Due to the very low signal strength of many ions, the CIMS data had been averaged over 20 s leading to enhance the reliability of the high-resolution analysis. This leads to an average desorption temperature difference  $\Delta T_{desorp} \approx 4^{\circ}C$  between two adjacent data points. To overcome this coarse  $T_{desorp}$  grid, we interpolated each factor's mass loading profile with a resolution of 100 sample points between two temperature steps to gain sufficient statistics for further analysis.

**15. Line 243. Missing close parenthesis**

We have added a closing parenthesis.

**16. Line 251. Out of curiosity, how was 0.3 nm chosen? This is approximately the length of 2 carbon-carbon bonds, so effectively a monolayer or thinner.**

The 0.3 nm was chosen to be close to one monolayer in the particle. In practice, we wanted to choose as small a value as possible for computational reasons that still makes physically sense.

**17. Line 256-258. This assertion is made a few times, but it's not clear to me if this is simply an assertion, or if it is also observed that the VD\_evap model suggests this to be true as well. This should be clarified/discussed**

The fact that the particles are produced in the same conditions and only evaporate in different conditions is an important aspect for interpreting these data and for modelling the evaporation. It means that when we determine  $VD_{evap}$  at t = 0 s from experiment in one relative humidity, we can use the same VD as the initial VD (at t = 0 s) for modelling the evaporation at different relative humidity.

The experiment set-up was specifically designed to assure that the starting composition of the particles were the same. We monitored the output from the OFR continuously with an AMS. FIGAERO-CIMS samples of the polydisperse Aerosol were taken on different experiment days. Both instruments showed very little variety in the particle composition with time. The monodisperse sample is then selected from this polydisperse distribution. The only difference will be the water content in the particles. But as we compare the organic mole fractions the

assumption still holds. The results from our  $VD_{evap}$  modelling do not contradict this aspect from the experimental set-up.

**18. Line 271. The mass spectra contains ions, not compounds. In any case, do I understand correctly that DeRieux et al provide a way to estimate Tg as a function of molecular formula? This isn't quite clear.**

The reviewer is correct here. We have clarified this in line 278 in the revised version of the manuscript.

This parametrization requires the number of carbon, oxygen and hydrogen atoms to calculate the  $T_{g}$ .

19. Line 285. What is the difference between Figures 1 and S2? I see that the contaminant and blank factors have been removed, and MD1 and has been split, but this is not all of the differences. For instance, in Figure S2 factor M1 coes to ~5000 signal, but only 2000 signal in Figure 1. The caption seem to imply these are the same data, but they don't look like it. Also, the desorption temperatures in Figure 1 go down to 0 C, which I don't think is correct.

Thank you for noticing this! The figure S2 wrongly shows the PMF factors for the experiments done in dry conditions. We have corrected the figure S2 to show all the PMF factors for the experiments done in high RH. The difference between Fig. 1 and Fig. S2 is that the latter shows the full 7- or 9- factor PMF results including also factors from filter background and from contamination (in the low O:C case). In Fig. 1, these background and contamination factors are removed and the decomposition factor (factors M/LD1 in Fig. S2) is split into two parts (factors M/LD1a/b in Fig. 1).

We have corrected also the issue where the desorption temperatures went down to 0 °C. In the plots where the desorption temperature went down to 0 °C, we had by accident included a data point below 23 °C that should not have been there.

20. Line 288. I think it's fine to point the reader to Buchholz for details, but a one or two sentence summary explanation is still necessary, I think, so the reader does not have to go to Buchholz unless they want the details. In other words, its fine to have a companion paper, but to be a separate paper, this paper still needs to be readable and understandable on its own.

The arguments for favoring a PMF solution over another need quite a bit of background information in this specific case. The details are explained in the companion paper. But to improve readability we added the following brief explanation in lines 294-297 in the revised version of the manuscript

We carefully investigated the  $Q/Q_{exp}$ , time series of scaled and unscaled residuals, and the ability of a PMF solution to capture the characteristic behavior of as many single ion thermograms as possible (see Buchholz 2019b for details). Based on this analysis, a 7-factor solution was chosen for the medium O:C cases and a 9-factor solution for the low O:C ones.

**21. Line 291. How was it determined that they were "clearly an artifact"?**

The dominant ions in these two factors were formic and lactic acid. Their signals in this sample were 10 times higher than in any other sample in the data set. While these two compounds do occur in SOA particles, they are also the most common contaminants for FIGAERO-CIMS

measurements. Our lab exhibits quite high gas-phase concentrations of formic acid due to e.g. using it as a calibrant. Filters exposed to room air will take up some formic acid from the gas phase. An elevated lactic acid signal typically stems from touching the FIGAERO inlet with bare hands during maintenance (which is not standard procedure but can happen). Normally, such contaminations on the filter or inlet would have been removed by the cleaning cycles performed each day. But clearly something must have gone wrong on that specific day.

The second piece of evidence can be derived from additional samples that were collected during the measurement campaign. We could not use that data due to malfunctions of the temperature sensors, i.e. no reliable  $T_{desorp}$  values were available. However, the total composition measurements (integrating over the full desorption cycle) are still valid and did not show such high formic and lactic acid values for low O:C, fresh, dry particles. Thus, we can conclude that the elevated values in the sample presented in the paper were indeed caused by a singular contamination of the filter or inlet. Normally, such a contaminated sample would be disregarded as "bad". But with the PMF analysis, we could identify the part of the signal affected by the contamination and remove it, thus salvaging the information from this sample.

**22. Line 295. This is a run-on sentence and should be split apart, probably at the first comma.**

We have split the sentence at the first comma.

**23. Line 299-301. I see why the authors split M/LD into two peaks, but this is a somewhat false dichomomty between the M/L peaks and the M/LD peaks - in reality they probably all contain some decomposition, it's just that for the other peaks it is a smooth enough transition as to appear monomodal.**

The key difference between the V- and D-type factors is that V-type factors exhibit 1 single peak (i.e. all compounds in it fall into a narrow volatility range), while the D-type factors have a flat or dual peak thermogram shape. This means that, yes, a V-type factor may contain thermal decomposition products. But then it will not contain the direct desorption part for that same ion. For the D-type factor, this is exactly what happens.

An example is shown below the factors M1 -M5 and MD1 for the ions  $[C_8H_{12}O_5 \cdot I]^-$  and  $[C_4H_2O_4 \cdot I]^-$ . The colored background indicates the direct desorption (grey) and thermal decomposition (red) temperature range that we applied for the splitting of MD1. Note how the first part of MD1 falls in the same  $T_{desorp}$  range as the V-type factors M1-M3. M5 falls in the thermal decomposition range where we also find the second part of MD1. So, factors M1-M3 are dominated by directly desorbing isomers of this composition. M5 is most likely dominated by thermal decomposition products. M4 may have contributions of both. This means that for all but 1 V-type factor the splitting of the integration area will have no effect as they are either almost completely in one or the other.

Figure AR2: Factor thermograms for the ion  $[C_8H_{12}O_5 \cdot I]^-$  (left) and  $[C_4H_2O_4 \cdot I]^-$  (right). The grey background indicates the direct desorption range and the red background the thermal decomposition range used for splitting MD1.

**24. Line 323. It would helpful to note here (based on the SI) that the selection of min, mean, or max evap time does not significantly impact the conclusions of this work.**

We have added the following to line 332 in the revised version of the manuscript.

The choice of sample evaporation time does not affect the conclusions we draw about the analysis presented in this section.

25. Figure 3. It might be helpful to include error bars. Sources of uncertainty on VD\_evap presumabely include min, mean, max evap time, and possible uncertainties in the LLEVAP model. Sources of uncertainty in VD\_PMF are harder to assess - definitely uncertainty in the T --> C\* conversion as discussed below, also theoretically PMF uncertainties but those are harder to understand in this context, and maybe other things?

Similarly, what is the benefit of grouping VD\_PMF in this way? It requires all the mass in one factor to be assigned into a bin based on T\_max. Why note just slice the thermogram by the relevant temperatures, and bin mass just based on evaporation temperature (without the need for PMF)? I guess that's actually an overall question - does PMF really improve modeling or understanding of volatility, or would slicing the thermogram into C\* bins just based on temperature yield basically the same conclusions?

We show the uncertainty associated with  $VD_{evap}$  in Figure S6. In our opinion the benefit of grouping the volatility distributions like we do in Figure 3 is that it allows us to compare  $VD_{evap}$  and  $VD_{PMF}$  on a qualitative level (answering for example questions like "which VD contains more non-volatile material"). This kind of analysis is hard to make solely from Figure 2 where the two distributions are plotted in a C\*-Xmole,dry plot. We mention the reasoning for this kind of analysis in lines 287-290.

Regarding the remark about the usefulness of PMF see our response to the second major comment.

26. Lines 391-392. It is reassuring that the desorption profile is usable in this way, but it is notable that uncertainty in the C\* of each factor seems to be on the order of 1 log unit based on Table 2. Such issues are typical in volatility experiments, but really suggest that this assumption of using T\_max to describe a PMF factor is highly uncertain. Unless the operator has some evapogram data to validate against (which is of course uncommon), it is not clear that such an assumption should be used, nor is it clear that uncertainty in the desorption temperature can feasibily be considered. This does not mean the present work is not valuable, because the assumptions being tested are being used by the community, but I think it actually should present more doubt or caution in applying the assumptions being tested.

We do put doubt on the ability of the PMF results to represent volatility when the volatility is calculated from the  $T_{max}$  in the discussion part of the manuscript (lines 514-517 in the revised version of the manuscript).

We have the added following also to the conclusions to lines 562-563 in the revised version of the manuscript.

[...]and it should be noted that deriving the volatilities based on only the  $T_{max}$  of PMF factors may not be sufficient for representing detailed SOA dynamics.

27. Line 506-507. Qualitatively I agree, but it's not clear that is quantitatively true - attempting to model evaporation using the PMF results alone (without optimization by comparison to an evapogram) yields moderately but not wholely successful results (Fig. 4). This is addressed below, but I think maybe this starting sentence needs to be tempered.

We agree and have added the following to lines 497-498 in the revised version of the manuscript to mark that the  $VD_{PMF}$  and  $VD_{PMF,dry}$  do not capture the evaporation dynamics quantitatively

Qualitatively,  $VD_{PMF}$  and  $VD_{PMF,dry}$  capture the evaporation dynamics well in all studied cases, although quantitatively there were discrepancies

**Comparing SOA volatility distributions derived from isothermal SOA particle evaporation data and FIGAERO-CIMS measurements**

Olli-Pekka Tikkanen1,2,3, Angela Buchholz1, Arttu Ylisirniö1, Siegfried Schobesberger1, Annele Virtanen1 and Taina Yli-Juuti1

1Department of Applied Physics, University of Eastern Finland, Kuopio, 70210, Finland
 2Department of Agricultural Sciences, University of Helsinki, Helsinki, 00790, Finland
 3Institute for Atmospheric and Earth System Research/Forest Sciences, Faculty of Agriculture and Forestry, University of Helsinki, Finland

10 *Correspondence to*: Olli-Pekka Tikkanen (o